# 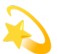 StarCoder: may the source be with you!

**Raymond Li**[2]  **Loubna Ben Allal**[1]  **Yangtian Zi**[4]  **Niklas Muennighoff**[1]  **Denis Kocetkov**[2]
**Chenghao Mou**[5]  **Marc Marone**[8]  **Christopher Akiki**[9,10]  **Jia Li**[5]  **Jenny Chim**[11]  **Qian Liu**[13]
**Evgenii Zheltonozhskii**[14]  **Terry Yue Zhuo**[15,16]  **Thomas Wang**[1]  **Olivier Dehaene**[1]  **Mishig**
**Davaadorj**[1]  **Joel Lamy-Poirier**[2]  **João Monteiro**[2]  **Oleh Shliazhko**[2]  **Nicolas Gontier**[2]
**Nicholas Meade**[6,17]  **Armel Zebaze**[1]  **Ming-Ho Yee**[4]  **Logesh Kumar Umapathi**[18]  **Jian Zhu**[19]
**Benjamin Lipkin**[20]  **Muhtasham Oblokulov**[21]  **Zhiruo Wang**[7]  **Rudra Murthy**[22]  **Jason**
**Stillerman**[23]  **Siva Sankalp Patel**[22]  **Dmitry Abulkhanov**[5]  **Marco Zocca**[24]  **Manan Dey**[25]
**Zhihan Zhang**[26]  **Nour Fahmy**[27]  **Urvashi Bhattacharyya**[28]  **Wenhao Yu**[26]  **Swayam Singh**[30]
**Sasha Luccioni**[1]  **Paulo Villegas**[31]  **Maxim Kunakov**[32]  **Fedor Zhdanov**[32]  **Manuel Romero**[5]
**Tony Lee**[33]  **Nadav Timor**[34]  **Jennifer Ding**[35]  **Claire Schlesinger**[4]
**Hailey Schoelkopf**[37]  **Jan Ebert**[38]  **Tri Dao**[33]  **Mayank Mishra**[22]  **Alex Gu**[20]  **Jennifer**
**Robinson**[3]  **Carolyn Jane Anderson**[36]  **Brendan Dolan-Gavitt**[29]  **Danish Contractor**[5]  **Siva**
**Reddy**[2,6]  **Daniel Fried**[7]  **Dzmitry Bahdanau**[2]  **Yacine Jernite**[1]  **Carlos Muñoz Ferrandis**[1]
**Sean Hughes**[3]  **Thomas Wolf**[1]  **Arjun Guha**[4,12]
**Leandro von Werra**[1,⋆]  **Harm de Vries**[2,⋆]

[1]Hugging Face  [2]ServiceNow Research  [3]ServiceNow  [4]Northeastern University  [5]Independent  [6]Mila
[7]Carnegie Mellon University  [8]Johns Hopkins University  [9]Leipzig University  [10]ScaDS.AI  [11]Queen Mary
University of London  [12]Roblox  [13]Sea AI Lab  [14]Technion – Israel Institute of Technology  [15]Monash
University  [16]CSIRO's Data61  [17]McGill University  [18]Saama AI Research Lab  [19]University of British
Columbia  [20]MIT  [21]Technical University of Munich  [22]IBM Research  [23]University of Vermont
[24]UnfoldML  [25]SAP  [26]University of Notre Dame  [27]Columbia University  [28]Discover Dollar Pvt Ltd
[29]NYU  [30]University of Allahabad  [31]Telefonica I+D  [32]Toloka  [33]Stanford University  [34]Weizmann
Institute of Science  [35]The Alan Turing Institute  [36]Wellesley College  [37]Eleuther AI
[38]Forschungszentrum Jülich

Corresponding authors (⋆) can be contacted at contact@bigcode-project.org

Reviewed on OpenReview: *https://openreview.net/forum?id=KoFOg41haE*

## Abstract

The BigCode community, an open-scientific collaboration working on the responsible development of Large Language Models for Code (Code LLMs), introduces StarCoder and StarCoderBase: 15.5B parameter models with 8K context length, infilling capabilities and fast large-batch inference enabled by multi-query attention. StarCoderBase is trained on 1 trillion tokens sourced from The Stack (Kocetkov et al., 2022), a large collection of permissively licensed GitHub repositories with inspection tools and an opt-out process. We fine-tuned StarCoderBase on 35B Python tokens, resulting in the creation of StarCoder. We perform the most comprehensive evaluation of Code LLMs to date and show that StarCoderBase outperforms every open Code LLM that supports multiple programming languages and matches or outperforms the OpenAI code-cushman-001 model. Furthermore, StarCoder outperforms every model that is fine-tuned on Python and still retains its performance on other programming languages. We take several important steps towards a safe open-access model release, including an improved PII redaction pipeline and a novel attribution tracing tool, and make the StarCoder models publicly available under a more commercially viable version of the Open Responsible AI Model license.

# 1 Introduction

Generative AI and large language models (LLMs; Brown et al., 2020; Chen et al., 2021; Chowdhery et al., 2022; Zhang et al., 2022; OpenAI, 2023a) are predicted to significantly impact the workforce in the coming years (Eloundou et al., 2023; Bommasani et al., 2021; World Economic Forum, 2023) by boosting worker productivity. LLMs trained on code (Code LLMs) have seen particularly fast adoption: Microsoft's Copilot has attracted over 1 million professional developers (Euronews, 2023) and GitHub reports that Copilot users rely on it to produce 35% of the code they write for some languages (Thompson, 2022). However, the development and use of LLMs has raised concerns of copyright, privacy, and openness.

Copyright concerns arise in many jurisdictions, including the U.S. and E.U., regarding the rights of content creators whose public data is used to train language models. It has been questioned whether machine learning models trained on such data fall under fair-use doctrine in the U.S. (Kuhn, 2022; Butterick, 2022; Rothchild & Rothchild, 2022), with fair use being most likely when the model generates novel content dissimilar to any copyrighted training data (Lemley & Casey, 2020; Levendowski, 2018). Henderson et al. (2023), therefore, suggest LLM developers should provide additional tools to ensure these models comply with current copyright laws. It is important to mention that these legal issues are not only the subject of scholarly debates: lawsuits have already been filed against GitHub Copilot (DOE 1 v. and GitHub, Inc., 2022) as well as Stable Diffusion (Andersen et al v. Stability AI et al, 2023).

Concerns about personal information led Italy to temporarily ban ChatGPT and launch an ongoing investigation into OpenAI's compliance with the E.U.'s General Data Protection Regulation (GDPR) (BBC, 2023). According to these regulations (European Council, 2018; Lomas, 2022), organizations that process personal information must have a valid legal basis. These laws could potentially affect LLM developers who gather vast amounts of public data from the internet, which may include personal information. Obtaining explicit consent from data creators is difficult at this scale, and it is uncertain whether other legal grounds exist for processing this personal information. Moreover, even with a valid legal basis, GDPR mandates that data processors inform individuals as to how their data is being processed and provide data access controls, such as the right to have data deleted or to modify erroneous data. This would require LLM providers to be transparent about the data they have collected and provide tooling for individuals to inspect their data and have the possibility to delete it.

The lack of transparency and openness surrounding the development processes of generative AI models has also raised concerns in the scientific community. Many models are closed-access to varying degrees: from being available only within the organization that developed them (Chowdhery et al., 2022; Hoffmann et al., 2022) to being accessible publicly through a paid API but with many details on their development process hidden (Brown et al., 2020; OpenAI, 2023a). While API access allows researchers to experiment with these models, it limits their ability to research LLM safety (Perez et al., 2022), inspect the models' inner workings (Olsson et al., 2022), and contribute to model improvements (Togelius & Yannakakis, 2023).

We use "open-access" to refer to models whose weights are public. Although other open-access models exist, the level of openness still varies across these projects; and some models with released weights have restrictions on model distribution (Touvron et al., 2023), or do not release their training datasets (Nijkamp et al., 2023; Zhang et al., 2022; Fried et al., 2022). Even in cases when models and training data are both released permissively (Raffel et al., 2020; Tay et al., 2022), external researchers typically do not have an opportunity to participate in guiding the development of industry-produced models. In contrast, other LLM development projects have taken a fully open approach which aims to allow for community inputs into model development, release training data, and enable external audits throughout the full development process (Solaiman, 2023). One example is the BigScience research workshop (BigScience Workshop, 2022), an open scientific collaboration (Akiki et al., 2022) comprising hundreds of researchers collaborating to release BLOOM, a multi-lingual LLM (Scao et al., 2022; Muennighoff et al., 2022). Similarly, EleutherAI, a grassroots-turned-nonprofit research initiative, has released open-access LLMs including GPT-NeoX (Black et al., 2022), GPT-J (Wang & Komatsuzaki, 2021), and Pythia (Biderman et al., 2023), as well as the associated training data (Gao et al., 2021a).

In this paper, we describe StarCoder and StarCoderBase, open-access code LLMs developed and released by the BigCode community, with a focus on respecting copyright, privacy, transparency, and community-driven

model development. The project is an open-scientific collaboration focusing on the responsible development of LLMs for code. It is co-stewarded by two industry research labs and comprises more than 600 members from diverse academic institutes and industry labs. The Stack (Kocetkov et al., 2022) is a publicly available pre-training dataset for Code LLMs with a transparent data governance framework. The Stack consists of 6.4 TB of permissively licensed source code in 384 programming languages, and includes 54 GB of GitHub issues and repository-level metadata in the v1.2 version of the dataset. The dataset comes with "Am I in The Stack", a governance tool for developers to check whether their source code is part of the dataset, and an opt-out process for those who wish to have their code removed from the dataset.

StarCoder and StarCoderBase are both 15.5B parameter models trained on permissively licensed data from The Stack. We trained StarCoderBase on 1 trillion tokens sourced from 80+ programming languages, GitHub issues, Git commits, and Jupyter notebooks. We fine-tuned StarCoderBase on another 35B Python tokens, leading to the StarCoder model. Both StarCoder models come with a novel combination of architectural features, such as an 8K token context length (Dao et al., 2022), infilling capabilities through Fill-in-the-Middle (FIM; Bavarian et al., 2022), and fast large-batch inference through Multi-Query-Attention (MQA; Shazeer, 2019). We present an extensive evaluation of the StarCoder models and release a demo along with an integrated attribution tool that can help users locate model generations that may have been copied from the training set. Overall, our contributions can be summarized as follows.

- We release StarCoderBase and StarCoder, open-access Code LLMs trained on 80+ programming languages that support a novel combination of capabilities and architectural features unavailable in other open Code LLMs.

- We perform the most comprehensive evaluation of Code LLMs to date using a diverse set of benchmarks (Lai et al., 2022; Cassano et al., 2023; Pearce et al., 2022; Fried et al., 2022; Yee & Guha, 2023; Austin et al., 2021; Chen et al., 2021; Ben Allal et al., 2022; Hendrycks et al., 2020; Reddy et al., 2019; Cobbe et al., 2021; Nadeem et al., 2021; Gehman et al., 2020; Liang et al., 2022), and show that:
  - *StarCoder outperforms every open LLM for code that supports multiple programming languages* (Nijkamp et al., 2023; Zheng et al., 2023);
  - *StarCoder matches or outperforms the OpenAI code-cushman-001 model*; and
  - When fine-tuned on Python, *StarCoder substantially outperforms existing LLMs that are also fine-tuned on Python.*

- We take important steps towards a safe open model release:
  - We release StarCoder under an *OpenRAIL-M license agreement*, which enables royalty-free access, use, and distribution of the model while embedding a set of use restrictions in identified critical scenarios. We have worked on a version of the license agreement that: (i) is more commercially viable for companies wishing to use and distribute the model and (ii) promotes transparency and understanding through the sharing of AI documentation such as model cards (Mitchell et al., 2019);
  - We incorporate a *new attribution tool into the VSCode demo that can help users detect and locate model generations that may have been copied from the training set.* This is achieved through a two-step process that involves a lightweight membership check followed by a search over a BM25 index (Section 9); and
  - *We have significantly improved the PII redaction pipeline by collecting a PII dataset containing 12,000 files with 22,950 annotated entities.* We fine-tuned our own encoder model (StarEncoder) on this dataset, resulting in a robust PII detection model (Section 4).

## 2 Related Work

**Language models** Early efforts to build large-scale language models used n-grams and simple smoothing techniques (Brants et al., 2007; Heafield et al., 2013; Buck et al., 2014). Other approaches applied various

types of neural networks architectures, such as feedforward networks (Bengio et al., 2000) and recurrent networks (Mikolov et al., 2010; Jozefowicz et al., 2016), to the language modeling task. The Transformer architecture (Vaswani et al., 2017) led to the development of highly scalable language models (Radford et al., 2019; Brown et al., 2020), which have shown a predictable relationship between language modeling loss and scaling factors such as the model size, number of training tokens, and compute budget (Kaplan et al., 2020; Hoffmann et al., 2022).

**Language Models for Code** Language models were initially applied to code by Hindle et al. (2012), but relied on n-gram models trained at comparatively small scale. Many neural architectures developed in NLP were also applied successfully to code, including encoder-only models for producing code representations (Feng et al., 2020; Kanade et al., 2020) and encoder-decoder models for translation, editing, summarization, and language-to-code tasks (Wang et al., 2021; Ahmad et al., 2021; Li et al., 2022). Decoder-only Transformer architectures have produced strong generative models of code, typically by training on mixtures of text and code from GitHub (Chen et al., 2021; Austin et al., 2021; Fried et al., 2022; Zheng et al., 2023; Nijkamp et al., 2023). Most of these models have not been fully open, but PolyCoder (Xu et al., 2022) and SantaCoder (Ben Allal et al., 2023) are notable exceptions and have both open models and training data. However, these models are relatively small (2.7B and 1.1B parameters, respectively) and are trained on less data ($< 300$GB of code) than we explore in this work.

**Closed-access LLMs** Several large tech companies have developed top-performing LLMs without releasing them. Examples include Google's PaLM (Chowdhery et al., 2022) and LaMDA (Thoppilan et al., 2022), DeepMind's Chinchilla (Hoffmann et al., 2022) and Gopher (Rae et al., 2021), and NVIDIA's Megatron-Turing NLG (Smith et al., 2022). OpenAI and other AI startups, including Cohere[1], Anthropic[2], and Aleph Alpha[3], offer LLMs as a paid API service. These companies did not release model weights nor provide comprehensive information on the methodology used to create these models. OpenAI has published several technical reports of the GPT family of models (Brown et al., 2020; Chen et al., 2021; OpenAI, 2023a), showcasing the capabilities of their models.

**Open-access LLMs** Numerous open-access LLMs have been released to the AI community, although they are generally not as strong as closed-access ones. In this paper, we use the term "open-access LLM" when the model weights are publicly available. We still note that there are significant differences between open-access models in how transparent they have been about the training data and filtering techniques. For instance, EleutherAI released GPT-NeoX-20B (Black et al., 2022) and GPT-J-6B (Wang & Komatsuzaki, 2021), as well as the dataset these models were trained on (Gao et al., 2021a). Google released UL2-20B (Tay et al., 2022), an encoder-decoder model trained on the publicly available C4 (Raffel et al., 2020). Tsinghua University released the weights of GLM-130B (Zeng et al., 2022), a Chinese-English LLM, and CodeGeeX-13B (Zheng et al., 2023), a LLM for coding applications, without releasing the training sets. Salesforce released CodeGen-Mono-16B (Nijkamp et al., 2023) without disclosing a proprietary Python dataset. Meta released the OPT (Zhang et al., 2022), LLaMA (Touvron et al., 2023), and InCoder models (Fried et al., 2022) under a non-commercial license and only provided high-level details about the data collection and filtering process.

## 3 Data Curation and Cleaning

This section describes how we processed the training data of StarCoderBase. We restrict the training set to The Stack v1.2 (Kocetkov et al., 2022), which exclusively contains data from permissively licensed[4] GitHub repositories. At the time of the data processing, 44 people opted out of The Stack. Below, we describe how we further cleaned the data by combining heuristic filtering and manual inspection.

---

[1] https://cohere.com/

[2] https://www.anthropic.com/

[3] https://www.aleph-alpha.com/

[4] See https://blueoakcouncil.org/ to learn more about permissive licenses and access a comprehensive collection of such licenses.

### 3.1 Programming Languages

**Selection of programming languages**  From the 358 programming languages in The Stack, we selected 86 languages. The assignment of data to programming languages was performed based solely on file extension (Kocetkov et al., 2022). We included all programming languages with more than 500 MB of data, as well as languages that were ranked in the top 50 on Githut 2.0 or the December 2022 TIOBE Index of programming language popularity. In addition, we included dialects of already selected programming languages (e.g., Racket and Scheme for Lisp). We excluded configuration languages (Nix, Puppet, etc.) and languages that are no longer actively supported (ActionScript). We also included data formats like JSON and YAML but limited its data volume (see "JSON and YAML" paragraph for details). The full list of selected programming languages can be found in Tables 1 and 2. Out of the languages present in MultiPL-E (Cassano et al., 2023), only D and Swift were not included in the training set. For D, language misclassification of the files led to less than 2MB of data in The Stack (Kocetkov et al., 2022). Swift was excluded from the final list of languages due to human error.

**Visual inspection**  We performed a visual inspection to ensure that we only retain data of high quality. To achieve this, we randomly selected 30,000 files from The Stack for each programming language, categorized them by extension, and kept a maximum of 1,000 files for each extension. We then reached out to our community for assistance with data inspection. We instructed the annotators to go through 50–100 files and confirm if the data appeared to be normal code written by humans, as opposed to text, data, or a single long line of autogenerated code. We also asked annotators to determine whether we should use our default alpha-numeric filter (which requires over 25% alpha-numeric symbols) and long-line filter (which requires lines to be less than 1,000 characters) for a given file extension. Eighteen community annotators evaluated 300 programming language extensions. After inspection, we excluded 36 extensions and eliminated the long-line filter for 27 extensions. The complete outcomes of the data inspection, including annotator remarks, can be found in this Google sheet.

**XML filter**  As we inspected the data, we noticed that certain extensions often consisted of XML files. For example, the `.sld` extension had more than 50% of its files in XML format. To address this, we implemented a simple XML filter that checked for the presence of "`<?xml version=`" within the first 100 characters of the file. This filter proved to be effective and produced few false positives. Hence, we applied it to all programming languages except for XSLT, which uses XML syntax.

**Alpha filter**  During our investigation, we discovered that certain extensions, such as MATLAB, contained numerous data files that frequently stored large tensors. To identify these files, we developed an alpha filter that removed files with fewer than 25% alphabetic characters. However, when we tested this filter on a small subset of data, we observed a high rate of false positives for certain programming languages, such as Assembly. To address this issue, we focused on the 25 extensions with the highest number of detections and manually verified whether or not the alpha filter should be applied.

**HTML**  We designed a custom HTML filter that targets excessive HTML boilerplate and links. We took into account the ratio of visible text in each file and only kept those files where the visible text makes up at least 20% of the HTML code and has a minimum length of 100 characters.

**JSON and YAML**  JSON and YAML files are naturally more data-heavy than other languages in The Stack. To remove most of the data files, we applied the following filters. For YAML, we kept files with 50–5000 characters, an average line length smaller than 100, a maximum line length smaller than 1000, and more than 50% alphabetic characters. These filters remove around 20% of the files and 90% of the volume. For JSON, we kept files with 50–5000 characters and more than 50% alphabetic characters, which removes around 70% of the files and 98% of the volume.

| Language | After dedup | | After filters and decont. | | Weight | Percentage |
|---|---|---|---|---|---|---|
| | Num. files | Volume (GB) | Num. files | Volume (GB) | | |
| ada | 31,291 | 0.30 | 30,934 | 0.26 | 0.26 | 0.034 |
| agda | 17,608 | 0.07 | 17,554 | 0.07 | 0.07 | 0.009 |
| alloy | 5,374 | 0.01 | 5,368 | 0.01 | 0.01 | 0.001 |
| antlr | 7,983 | 0.05 | 7,917 | 0.05 | 0.05 | 0.007 |
| applescript | 4,906 | 0.01 | 4,737 | 0.01 | 0.01 | 0.001 |
| assembly | 248,396 | 1.58 | 247,919 | 1.56 | 1.56 | 0.203 |
| augeas | 195 | 0.00 | 180 | 0.00 | 0.00 | 0 |
| awk | 10,430 | 0.02 | 10,289 | 0.02 | 0.02 | 0.003 |
| batchfile | 252,514 | 0.29 | 239,568 | 0.23 | 0.23 | 0.03 |
| bluespec | 5,940 | 0.03 | 5,928 | 0.03 | 0.03 | 0.004 |
| c | 8,625,559 | 57.43 | 8,536,791 | 53.89 | 53.89 | 7.027 |
| c-sharp | 10,839,399 | 46.29 | 10,801,285 | 44.66 | 44.66 | 5.823 |
| clojure | 126,191 | 0.49 | 125,163 | 0.46 | 0.46 | 0.06 |
| cmake | 186,517 | 0.45 | 186,375 | 0.45 | 0.45 | 0.059 |
| coffeescript | 227,889 | 0.69 | 226,209 | 0.64 | 0.64 | 0.083 |
| common-lisp | 101,370 | 1.68 | 98,733 | 1.40 | 1.40 | 0.183 |
| cpp | 6,377,914 | 50.89 | 6,353,527 | 48.92 | 48.92 | 6.379 |
| css | 2,994,829 | 22.61 | 2,721,616 | 11.93 | 3.00 | 0.391 |
| cuda | 58,355 | 0.59 | 58,151 | 0.56 | 0.56 | 0.073 |
| dart | 932,583 | 3.86 | 928,415 | 3.66 | 3.66 | 0.477 |
| dockerfile | 572,186 | 0.42 | 571,506 | 0.42 | 0.42 | 0.055 |
| elixir | 282,110 | 0.74 | 281,016 | 0.71 | 0.71 | 0.093 |
| elm | 62,861 | 0.34 | 62,033 | 0.30 | 0.30 | 0.039 |
| emacs-lisp | 54,768 | 0.43 | 52,838 | 0.41 | 0.41 | 0.053 |
| erlang | 99,368 | 0.73 | 98,447 | 0.70 | 0.70 | 0.091 |
| f-sharp | 127,161 | 0.90 | 124,066 | 0.61 | 0.61 | 0.08 |
| fortran | 165,446 | 1.84 | 158,792 | 1.78 | 1.78 | 0.232 |
| glsl | 175,576 | 0.57 | 167,701 | 0.40 | 0.40 | 0.052 |
| go | 4,730,461 | 25.74 | 4,700,526 | 23.78 | 23.78 | 3.101 |
| groovy | 251,627 | 0.94 | 250,834 | 0.91 | 0.91 | 0.119 |
| haskell | 544,969 | 2.36 | 541,454 | 2.23 | 2.23 | 0.291 |
| html | 9,533,367 | 146.76 | 3,299,965 | 29.36 | 29.36 | 3.828 |
| idris | 8,060 | 0.03 | 8,042 | 0.03 | 0.03 | 0.004 |
| isabelle | 5,086 | 0.09 | 5,001 | 0.08 | 0.08 | 0.01 |
| java | 20,151,565 | 89.30 | 20,071,773 | 86.94 | 86.94 | 11.336 |
| java-server-pages | 214,133 | 1.03 | 210,816 | 0.98 | 0.98 | 0.128 |
| javascript | 21,108,587 | 141.65 | 19,544,285 | 64.71 | 64.71 | 8.437 |
| json | 17,012,912 | 338.34 | 4,751,547 | 5.62 | 1.00 | 0.13 |
| julia | 298,672 | 1.54 | 295,364 | 1.31 | 1.31 | 0.171 |
| kotlin | 2,242,771 | 5.77 | 2,239,354 | 5.68 | 5.68 | 0.741 |
| lean | 16,891 | 0.10 | 16,870 | 0.09 | 0.09 | 0.012 |
| literate-agda | 523 | 0.01 | 523 | 0.01 | 0.01 | 0.001 |
| literate-coffeescript | 1,138 | 0.01 | 1,133 | 0.01 | 0.01 | 0.001 |
| literate-haskell | 6,135 | 0.05 | 6,104 | 0.05 | 0.05 | 0.007 |
| lua | 558,861 | 3.28 | 549,459 | 2.87 | 2.87 | 0.374 |
| makefile | 661,424 | 1.49 | 657,349 | 1.31 | 1.31 | 0.171 |
| maple | 1,259 | 0.01 | 1,152 | 0.01 | 0.01 | 0.001 |
| markdown | 21,045,171 | 75.25 | 21,029,287 | 74.93 | 74.93 | 9.77 |
| mathematica | 26,895 | 1.72 | 22,653 | 1.25 | 1.25 | 0.163 |
| matlab | 967 | 0.04 | 93 | 0.00 | 0.00 | 0 |

Table 1: Overview of the training data for StarCoder. For the selected programming languages, we show the number of files and data volume after near-deduplication, as well as after filtering. See also Table 2.

| Language | After dedup | | After filters and decont. | | Weight | Percentage |
|---|---|---|---|---|---|---|
| | Num. files | Volume (GB) | Num. files | Volume (GB) | | |
| ocaml | 159,734 | 1.11 | 158,356 | 1.03 | 1.03 | 0.134 |
| pascal | 118,675 | 1.71 | 110,981 | 1.68 | 1.68 | 0.219 |
| perl | 392,108 | 2.63 | 365,491 | 2.23 | 2.23 | 0.291 |
| php | 15,904,518 | 66.84 | 15,683,017 | 60.89 | 60.89 | 7.939 |
| powershell | 271,487 | 1.25 | 267,627 | 1.12 | 1.12 | 0.146 |
| prolog | 1,023 | 0.01 | 968 | 0.01 | 0.01 | 0.001 |
| protocol-buffer | 98,246 | 0.44 | 97,167 | 0.31 | 0.31 | 0.04 |
| python | 12,962,249 | 64.30 | 12,866,649 | 60.40 | 60.40 | 7.875 |
| r | 39,194 | 0.30 | 39,042 | 0.30 | 0.30 | 0.039 |
| racket | 4,201 | 0.04 | 3,688 | 0.03 | 0.03 | 0.004 |
| restructuredtext | 905,679 | 3.42 | 896,880 | 3.32 | 3.32 | 0.433 |
| rmarkdown | 5,389 | 0.06 | 5,386 | 0.06 | 0.06 | 0.008 |
| ruby | 3,405,374 | 7.14 | 3,390,320 | 6.81 | 6.81 | 0.888 |
| rust | 1,386,585 | 9.53 | 1,380,468 | 9.11 | 9.11 | 1.188 |
| sas | 9,772 | 0.13 | 9,226 | 0.12 | 0.12 | 0.016 |
| scala | 1,362,426 | 4.86 | 1,355,788 | 4.69 | 4.69 | 0.612 |
| scheme | 44,261 | 0.30 | 41,890 | 0.20 | 0.20 | 0.026 |
| shell | 2,236,434 | 3.38 | 2,206,327 | 3.09 | 3.09 | 0.403 |
| smalltalk | 592,999 | 0.74 | 587,748 | 0.58 | 0.58 | 0.076 |
| solidity | 164,242 | 1.21 | 153,194 | 0.85 | 0.85 | 0.111 |
| sparql | 14,173 | 0.04 | 13,716 | 0.04 | 0.04 | 0.005 |
| sql | 994,019 | 12.22 | 975,420 | 11.09 | 11.09 | 1.446 |
| stan | 5,441 | 0.01 | 5,429 | 0.01 | 0.01 | 0.001 |
| standard-ml | 48,995 | 0.52 | 19,630 | 0.19 | 0.19 | 0.025 |
| stata | 31,282 | 0.41 | 24,208 | 0.33 | 0.33 | 0.043 |
| systemverilog | 46,915 | 0.41 | 46,270 | 0.39 | 0.39 | 0.051 |
| tcl | 50,579 | 0.40 | 49,335 | 0.35 | 0.35 | 0.046 |
| tcsh | 4,911 | 0.02 | 4,806 | 0.02 | 0.02 | 0.003 |
| tex | 547,888 | 5.44 | 522,778 | 5.20 | 5.20 | 0.678 |
| thrift | 4,663 | 0.01 | 4,661 | 0.01 | 0.01 | 0.001 |
| typescript | 10,637,070 | 28.82 | 10,547,331 | 26.52 | 26.52 | 3.458 |
| verilog | 77 | 0.001 | 75 | 0.001 | 0.001 | 0 |
| vhdl | 60,027 | 1.12 | 58,208 | 0.94 | 0.94 | 0.123 |
| visual-basic | 163,291 | 1.49 | 161,239 | 1.42 | 1.42 | 0.185 |
| xslt | 43,095 | 0.56 | 6,513 | 0.05 | 0.05 | 0.007 |
| yacc | 25,775 | 0.41 | 7,451 | 0.11 | 0.11 | 0.014 |
| yaml | 5,282,081 | 28.36 | 3,995,948 | 3.76 | 1.00 | 0.13 |
| zig | 15,913 | 0.18 | 15,850 | 0.18 | 0.18 | 0.023 |
| GitHub issues | | | ∼ 30,900,000 | 54.40 | 54.40 | 7.093 |
| Git commits | | | 7,674,345 | 64.00 | 32.00 | 4.172 |
| notebook scripts | | | 914,000 | 7.12 | 7.12 | 0.928 |
| notebook structured | | | 668,743 | 6.00 | 6.00 | 0.782 |
| | | | 305,929,658 | 815.68 | 799.37 | 100 |

Table 2: Overview of the training data for StarCoder. For the selected programming languages, we show the number of files and data volume after near-deduplication, as well as after filtering. See also Table 1.

## 3.2   Jupyter notebooks

All Jupyter notebooks were retrieved from the Stack. We transformed Jupyter notebooks into two different datasets: Jupyter – scripts and Jupyter – structured.

| Language | Num files | Percentage |
|---|---|---|
| python | 1,392,432 | 97.170 |
| julia | 16,730 | 1.167 |
| r | 11,034 | 0.77 |
| scala | 1,899 | 0.133 |
| bash | 1,441 | 0.101 |
| java | 1,319 | 0.092 |
| q-sharp | 1,273 | 0.089 |
| cpp | 1,081 | 0.075 |
| c-sharp | 1,048 | 0.073 |
| matlab | 908 | 0.063 |
| powershell | 769 | 0.054 |
| javascript | 592 | 0.041 |
| haskell | 535 | 0.037 |
| scheme | 484 | 0.034 |
| groovy | 432 | 0.03 |
| f-sharp | 385 | 0.027 |
| ocaml | 279 | 0.019 |
| rust | 134 | 0.009 |
| clojure | 96 | 0.007 |
| typescript | 72 | 0.005 |
| maxima | 31 | 0.002 |
| coconut | 6 | 0 |
| markdown | 5 | 0 |
| wolfram language | 4 | 0 |
| tcl | 3 | 0 |
| Total | 1,432,992 | 100 |

Table 3: Overview of the initially collected Jupyter scripts, with the number of files and the percentage.

**Jupyter − scripts**  We utilize Jupytext[5] to convert notebooks to scripts. It is an actively maintained software that currently supports 31 programming languages. To initiate the conversion process, Jupytext requires the identification of the specific programming languages within each notebook. We extracted this information from the metadata of each respective notebook. However, more than 30,000 notebooks lacked any programming language information, making it difficult to convert them to the script format. To address this issue, we incorporated the use of Guesslang,[6] an open-source library that employs machine learning techniques to identify the programming languages of source code. By applying a probability threshold greater than or equal to 0.5, we successfully reduced the number of unidentified notebooks to 6,400 using Guesslang. Ultimately, we amassed 1,432,992 scripts through the utilization of Jupytext. The distribution of programming languages among these scripts is presented in Table 3. We evaluated language coverage by randomly selecting 100 files from the transformed scripts, ensuring that all programming languages were represented within this sample.

**Jupyter − structured**  To create this dataset, we first filtered out notebooks that did not contain any Python code or Markdown text. The information on the programming language in the metadata of each notebook was used as the criterion to filter out non-Python notebooks. Only notebooks explicitly marked as 'Python' in the metadata were kept. Then for each notebook, consecutive Markdown blocks or code blocks were merged into a large Markdown or code block respectively. Eventually, we ended up with consecutive code-text pairs in temporal order grouped by each notebook. In general, each Jupyter code-text pair contained the Markdown text immediately preceding the code block and the Python code, which forms a natural

---

[5]https://jupytext.readthedocs.io/
[6]https://guesslang.readthedocs.io/

instruction pair. We also included the formatted output of a code block if the output cell was non-empty; otherwise, it was marked by a special `<empty_output>` token. If consecutive code blocks have multiple output cells before merging, we only retain the output of the last code block. After these preprocessing steps, we ended up with 1,045,605 structured Jupyter notebooks.

### 3.3 GitHub issues

We used natural language conversations from GitHub issues and pull requests, which were collected as a component of The Stack v1.2. Each conversation consists of a series of events with actions, such as opening the issue, creating a comment, or closing the issue. Each event includes the author's username, a message, an action, and a creation date. We filtered this data as follows: 1) First, we removed auto-generated text when users replied to issues via email. See Appendix A for the regular expression we used. We also deleted issues with a short message (less than 200 characters) and truncated long comments in the middle to a maximum of 100 lines while retaining the last 20 lines. This removed 18% of the volume. 2) Next, we excluded comments from bots. To do so, we searched for bot keywords in the username of the comment's author (for more information, see Appendix A). This step eliminates 17% of the total events and results in 14.7% of the issues being emptied. We have observed that bot-generated issues tend to be lengthy and contain numerous logs and links. 3) We used the number of users engaged in the conversation as an indicator of quality. Our criterion was to include conversations that have two or more users. However, we also preserved conversations that involved a single user if the total text within comments was less than 7,000 characters (96th percentile). Additionally, we excluded issues authored by a single user if they contained more than ten events, as they tended to be of poor quality or originate from overlooked bots. By implementing these filters, we removed an additional 14% of issues. 4) Finally, we used a model from the *fasttext* library[7] to filter out non-English issues. This step was necessary to enable accurate redaction of names using a PII detection model (see Section 4.3).

Lastly, we would like to point out that we anonymized the usernames in the conversations by replacing them with a participant counter within the conversation. See more details in Section 4.3 and 5.1.

### 3.4 Git commits

The Git commit data was gathered from BigQuery[8] and includes only single-file commits of repositories with the same licenses and file extension as used in The Stack (Kocetkov et al., 2022). We removed all repositories from users that opted out of The Stack. The raw dataset is around 4 TB in size. We sampled 50% of the files and filtered the remaining data with heuristics to build a high-quality dataset. We list and describe all filters in Table 4.

The number of line changes in a commit can be very low compared to the file size. To avoid spending too much compute budget on learning to copy the file content, we only used the full file 20% of the time, and for the remaining 80%, sampled a window between 0 and 32 lines around the first and last changed line. The resulting dataset contains 64 GB of commit data.

### 3.5 Deduplication

We followed the deduplication pipeline from Ben Allal et al. (2023), which consists of calculating the MinHashes (Broder, 2000) of all source code files, followed by Locally Sensitive Hashing (LSH) to map similar code files to the same bucket. We used 5-grams and a Jaccard similarity of 0.7. See this blogpost for more details regarding the pipeline.

We applied this near-deduplication process to all programming languages and the Jupyter notebooks. However, due to time constraints, we could not apply this procedure to Git commits. Additionally, we deemed it unlikely to discover duplicates in Github issues, so we didn't apply the process to them.

---

[7]The lid.176.bin version of this language identification model: https://fasttext.cc/docs/en/language-identification.html
[8]https://cloud.google.com/bigquery/public-data/

| Description | Details |
|---|---|
| Maximum characters | Remove code files with >100k characters. |
| Small changes | Subsample changes with $\leq 2$ lines with 50% probability. |
| Long-range refactorings | Subsample changes spanning $\geq 200$ lines with 10% probability. |
| Empty commit message | Remove commits with empty commit subject. |
| Automatic commits | Remove commits that either contain or are equal to a list of stop words. |
| Hash messages | Remove commits with whitespace-separated words-to-character ratio >20. |
| Data files | Subsample data formats (JSON, YAML, XML, HTML) with 50% probability. |

Table 4: Git commit filters.

## 3.6 Weighting of data sources

There were several discussions within the community about whether to up-sample or down-sample certain programming languages, as the amount of compute budget allocated to a data source in a given language can significantly affect the model's performance in that language. However, we realized that the largest amount of available data comes from popular programming languages and would, therefore, benefit a larger group of end-users. Moreover, after the deduplication process, we found that several high-resource programming languages, such as C, C++, C#, Java, Javascript, Python, and PHP, had a similar amount of data ranging from 44–87 GB. This further reinforced our belief that we did not need to drastically re-weigh the existing data distribution. Thus, in this work, we followed the natural distribution of data during training and sampled data sources proportionally to their volume. However, we did make an exception for JSON, YAML, and CSS, as we only want the LLM to learn the data format without wasting compute resources on memorizing the data in such files. For that reason, we re-weighed the volume of the data source to 1 GB for JSON and YAML and 3GB for CSS.

## 4 PII redaction

This section outlines our efforts to remove Personally Identifiable Information (PII) from the training data. In Section 4.1, we first describe how we collected a large set of PII annotations. We used these annotations to explore various techniques to train a PII detection model in Section 4.3, building on top of the encoder model we developed in Section 4.2.

## 4.1 Data collection

We utilized the Toloka platform[9] to engage 1,399 crowd-workers from 35 countries in annotating a dataset for PII in source code. On average, participants completed 206 tasks, earned about $27, and worked 3.1 hours. Our goal was to identify PII in various forms, such as names, usernames, emails, IP addresses, keys, passwords, and IDs. To ensure that crowd-workers received fair compensation, we established an hourly pay rate of $7.30, taking into consideration different minimum wage rates across countries and their corresponding purchasing power. We limited annotation eligibility to countries where the hourly pay rate of $7.30 was equivalent to the highest minimum wage in the US ($16.50) in terms of purchasing power parity. A complete list of countries that participated in the annotation can be found in Table B.1 of Appendix B. Crowd workers in Toloka can do tasks whenever or wherever; there is no obligation to complete a certain task or spend a fixed amount of time on it. Thus, they utilize free choice when working on the tasks. Out of 1,399 crowd workers, 695 filled a survey on task quality, and 519 completed the survey. The average score for the question asking whether the participant would like to contribute to another project like this is 4.92 on a scale 1–5.

---

[9]https://toloka.ai/

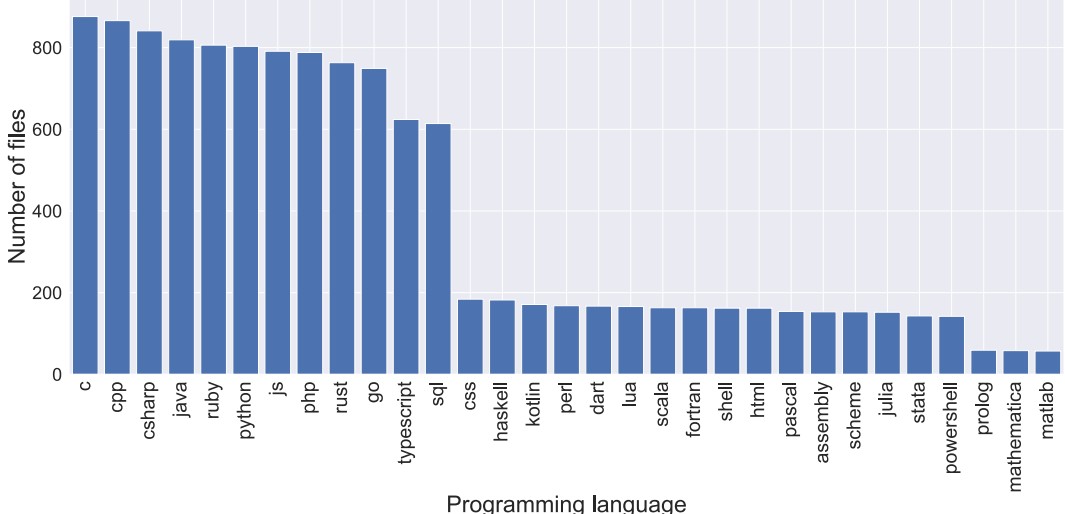

Figure 1: Distribution of programming languages in the annotated PII dataset.

The dataset comprises 12,000 files, each containing approximately 50 lines of code written in 31 programming languages. Figure 1 shows the distribution of programming languages in the dataset. To increase the representation of rare PII types, such as keys and IP addresses, 7,100 files were pre-filtered from a larger sample. We utilized the `detect-secrets` tool[10] with all default plugins activated, along with the regular expressions by Ben Allal et al. (2023) for detecting emails, IPv4 and IPv6 addresses. To prevent biasing the annotation too much towards these detection tools, the remaining 5,100 files were randomly selected from the dataset without pre-filtering.

During annotation, we differentiated between various types of PII based on the specific context in which it appeared. Specifically, we distinguished whether the PII was present in the code's license header, was used as a placeholder, or constituted confidential data. This categorization was necessary because the PII in license headers is usually provided voluntarily by authors for code attribution and may not require masking. Similarly, placeholders are not real secrets and do not need to be masked. We applied this categorization to names, emails, and usernames. See Table 5 for an overview of all PII entities.

The annotators detected a total of 22,950 PII entities in the dataset. To evaluate the quality of the dataset, we manually inspected 300 files that contained various PII types and calculated the recall and precision for each type, as shown in Table 5. We found that annotating secret IDs was particularly challenging, as the annotators tended to produce many false positives and negatives. As a result, we decided to exclude this category from the PII detection model training.

## 4.2 StarEncoder

As part of our PII detection efforts, we trained an encoder-only model (i.e., bi-directionally self-attentive Transformers) that can be efficiently fine-tuned for both code- and text-related tasks. We used the Masked Language Modelling (MLM) and Next Sentence Prediction (NSP) objectives from BERT (Devlin et al., 2019; Liu et al., 2019) and predicted masked-out tokens from an input sentence and whether a pair of sentences occur as neighbors in a document.

We separate code snippets in the input as follows: `[CLS]` Snippet-1 `[SEP]` Snippet-2, where the two code snippets are selected randomly, either from the same source file or from two distinct documents. For the MLM loss, we mask tokens in the input independently with an probability of 15%. For the NSP loss, we use a linear classifier applied to the representation output at the `[CLS]` token. We train for 100,000 steps with a global batch size of 4,096 sequences of a maximum length of 1,024 so that approximately 400B tokens are

---

[10]https://github.com/Yelp/detect-secrets

| PII type | Count | Recall | Precision |
|---|---|---|---|
| IP_ADDRESS | 2526 | 85% | 97% |
| KEY | 308 | 91% | 78% |
| PASSWORD | 598 | 91% | 86% |
| ID | 1702 | 53% | 51% |
| EMAIL | 5470 | 99% | 97% |
| EMAIL_EXAMPLE | 1407 | | |
| EMAIL_LICENSE | 3141 | | |
| NAME | 2477 | 89% | 94% |
| NAME_EXAMPLE | 318 | | |
| NAME_LICENSE | 3105 | | |
| USERNAME | 780 | 74% | 86% |
| USERNAME_EXAMPLE | 328 | | |
| USERNAME_LICENSE | 503 | | |
| AMBIGUOUS | 287 | | |

Table 5: Overview of the PII types and the number of collected annotations. We investigate the annotation quality by reporting the precision and recall of a manual inspection on 300 files. Each subcategory was mapped back to its corresponding PII type for the inspection.

| Hyperparameter | Value |
|---|---|
| Hidden size | 768 |
| Intermediate size | 3072 |
| Max. position embeddings | 1024 |
| Num. of attention heads | 12 |
| Num. of hidden layers | 12 |
| Attention | Multi-head |
| Num. of parameters | ≈125M |

Table 6: Model architecture of StarEncoder.

observed. This takes roughly two days using 64 NVIDIA A100 GPUs. Details about the model architecture are reported in Table 6.

### 4.3 PII detection model

We fine-tuned StarEncoder on the annotated PII dataset for the Named Entity Recognition (NER) task. We added a linear layer as a token classification head on top of the model, with 6 target classes: names, emails, keys, passwords, IP addresses, and usernames. We excluded IDs due to low annotation quality and did not differentiate between the categorization of PII entities (license headers, placeholders) because of the model's poor performance in distinguishing them. We split the dataset into a training set of 7,878 examples and a test set of 4,000 examples, ensuring that both splits have a balanced representation of the different PII types. See Table 7. We make the training and evaluation splits available under gated access at `https://hf.co/BigCode`.

**Fine-tuning baseline** We fine-tune StarEncoder on the PII training set, and 400 annotated files from Ben Allal et al. (2023). We achieve F1 scores of more than 90% on names, emails, and IP addresses and 73.39% on passwords. The model's performance is comparatively low on keys and usernames, with F1 scores of only 56.66% and 59.39%, respectively. We attribute the low performance on keys to the limited number of labels for this type of PII, as only 308 instances were available. For usernames, we observed the model often confused them with decorators and values in paths. This is most likely because we annotated usernames inside links for social media platforms.

| Entity type | Train | Test |
|---|---|---|
| EMAIL | 4721 | 1742 |
| NAME | 3847 | 1298 |
| IP_ADDRESS | 1941 | 521 |
| USERNAME | 1320 | 346 |
| PASSWORD | 390 | 148 |
| KEY | 171 | 118 |

Table 7: Train-test split of the annotated PII dataset.

| Method | Email address | | | IP address | | | Key | | |
|---|---|---|---|---|---|---|---|---|---|
| | Prec. | Recall | F1 | Prec. | Recall | F1 | Prec. | Recall | F1 |
| Regex | 96.20% | 97.47% | 96.83% | 71.29% | 87.71% | 78.65% | 3.62% | 49.15% | 6.74% |
| NER | 94.01% | 98.10% | 96.01% | 88.95% | **94.43%** | 91.61% | 60.37% | 53.38% | 56.66% |
| + pseudo labels | **97.73%** | **98.94%** | **98.15%** | **90.10%** | 93.86% | **91.94%** | **62.38%** | **80.81%** | **70.41%** |

Table 8: Comparing PII detection performance: Regular Expressions, NER Pipeline with Annotated Data, and NER Pipeline with Annotated Data + Pseudo-Labels

**Pseudo-labels** To improve the detection of key and password entities, we employed a pseudo-labeling technique as described by Lee (2013). This method involves training a model on a small set of labeled data and subsequently generating predictions for a larger set of unlabeled data. Specifically, we annotated 18,000 files using an ensemble of two encoder models, which were fine-tuned on the 400-file PII dataset from Ben Allal et al. (2023). To identify reliable pseudo-labels, we calculated the average probability logits from our models and applied filtering criteria. Specifically, we set a minimum threshold of 0.5 for all entities, except for names and usernames, for which we used a higher threshold of 0.6. However, upon reviewing the results, we found a significant number of false positives for keys and passwords. As a result, we decided to only retain entities that were preceded by a trigger word, such as `key`, `auth`, or `pwd`, within the preceding 100 characters. Training on this synthetic dataset before fine-tuning on the annotated one yielded superior results for all PII categories, as demonstrated in Tables 8 and 9. Only the performance for detecting usernames did not show significant improvement, so we decided to exclude it from the PII redaction process.

**Comparison against regex baseline** We compared our PII detection models against the regular expressions (regexes) employed in Ben Allal et al. (2023). The regexes only support the detection of emails, IP addresses, and keys. Note that we enhanced the email regex, as explained in the Appendix, to address false positives we found during the evaluation on this benchmark. This modification boosted the F1 score of the regex from 81.8% to 96.83%. Nevertheless, our PII detection models still surpassed the regex approach in detecting all three entities, as shown in Table 8. We note that the performance difference was especially large on keys and found that the `detect-secrets` tool generated many false positives, especially in specific programming languages like Go and C-sharp that weren't well represented in the regex evaluation. Consequently, the overall precision of the tool was below 4%.

**Post-processing** Before applying the best PII detection model to the full dataset, we observed a couple of frequent detection errors. We added the following post-processing techniques to reduce the number of false positives:

- Ignore secrets with fewer than 4 characters.

- Detect full names only by requiring at least one space within the name.

| Method | Name | | | Username | | | Password | | |
|---|---|---|---|---|---|---|---|---|---|
| | **Prec.** | **Recall** | **F1** | **Prec.** | **Recall** | **F1** | **Prec.** | **Recall** | **F1** |
| NER | 83.66% | 95.52% | 89.19% | 48.93% | **75.55%** | 59.39% | 59.16% | **96.62%** | 73.39% |
| + pseudo labels | **86.45%** | **97.38%** | **91.59%** | **52.20%** | 74.81% | **61.49%** | **70.94%** | 95.96% | **81.57%** |

Table 9: Comparison of PII detection performance: NER Pipeline with Annotated Data vs. Annotated Data + Pseudo-Labels

- Ignore detected keys with fewer than 9 characters or that are not gibberish using a `gibberish-detector`.[11]

- Ignore IP addresses that aren't valid or are private (non-Internet facing) using the `ipaddress` python package. We also ignore IP addresses from popular DNS servers. We use the same list as in Ben Allal et al. (2023).

**PII placeholders**   We replaced the detected PII entities with the following tokens:

`<NAME>, <EMAIL>, <KEY>, <PASSWORD>`

To mask IP addresses, we randomly selected an IP address from 5 synthetic, private, non-internet-facing IP addresses of the same type that can be found in Appendix C.

**Github issues**   We already employed a regex approach to detect keys, IP addresses, and emails in the Github issues, so we only used the PII detection model to redact names. We anonymized the usernames of the authors by replacing them with a participant counter within the conversation, e.g. username_1 to refer to second participant (see Section 5.1 for formatting details). We prepend these pseudonyms to the beginning of each comment such that we preserve the speaker identity of the author. In addition, we redact all mentions of these usernames in the messages. Note that we only mask the usernames of active participants in the conversation and mentions of non-participating users are not anonymized.

**Compute resources**   We used the PII detection model to identify PII across all programming languages in the training dataset, including GitHub issues (names only), Git commits, and Jupyter notebooks. The total dataset amounts to 815 GB in size. We ran inference on multiple NVIDIA A100 80 GB GPUs, which required 800 GPU-hours.

## 5   Model training

This section presents information on the training process of the StarCoder models. Before we proceed, we first clarify the differences between the two models:

**StarCoderBase** is the first model trained on 1 trillion tokens sourced from the curated dataset described in Section 3.

**StarCoder** is the fine-tuned version of StarCoderBase, trained on another 35B Python tokens (roughly 2 epochs).

Throughout the following, we show how we formatted the training data (Section 5.1), decontaminated the training data (Section 5.2), and provide details regarding the tokenizer (Section 5.3), the model architecture (Section 5.4), the training process (Section 5.5), multi-node GPU setup (Section 5.6), and CO2 emissions (Section 5.7).

---

[11]https://github.com/domanchi/gibberish-detector

### 5.1 Data formatting

We present the formatting guidelines for each of the data sources below. We provide the templates below in which <token> refers to a sentinel token, and metadata and data refer to placeholders for data fields, respectively.

**Code**  We prepend the repository name, file name, and the number of stars to the context of the code file. To not overfit on the exact number of stars, we categorized GitHub stars into five buckets: 0, 1–10, 10–100, 100–1000, 1000+. To enable the model to operate without this metadata during inference, we prefixed the repository name, filename, and stars independently at random, each with a probability of 0.2.

```
<reponame>reponame<filename>filename<gh_stars>stars\ncode<|endoftext|>
```

To the source code in this template (i.e. code), we apply the **fill-in-the-middle transformation** (FIM; Bavarian et al., 2022). More precisely, we apply FIM at the character-level to the source code files with a FIM-rate of 0.5, and use PSM mode with probability .5 and SPMv2 mode with probability .5.

**Issues**  We use sentinel tokens to mark the opening of an issue and subsequently include its title. We separate the sequence of comments by a <issue_comment> token and include a anonymized speaker identifier before the comment. Specifically, we refer to authors by their participant counter within the conversation, e.g. username_1 to refer to second participant in the issue. To distinguish between the different turns, we use comment1, id1 to refer to the second comment and its anonymized speaker id, respectively.

```
<issue_start>Title: title\nusername_id0:comment0<issue_comment>username_id1:comment1
... <issue_closed (optional)><|endoftext|>
```

**Jupyter − scripts**  Jupyter scripts were formatted in the same manner as code.

**Jupyter − structured**  Parsed Jupyter notebooks come in chains of text, code, and outputs, and we separated them with sentinel tokens. Note that we use text2, code2, output2 to refer to the 3rd triplet in the notebook.

```
<jupyter_start><jupyter_text>text0<jupyter_code>code0
<jupyter_output>output0<jupyter_text> ... <|endoftext|>
```

**Git commits**  We separate the code before the commit, the commit message, and the code after the commit with sentinel tokens. As explained in Section 3.4, we use the full files with 20% probability and otherwise use a small window (0-32 lines) around the changed lines.

```
<commit_before>code_before<commit_msg>message<commit_after>code_after<|endoftext|>
```

We summarize all sentinel tokens in Table 10.

### 5.2 Training data decontamination

The code training data was decontaminated by removing files that contained docstrings or solutions from HumanEval and MBPP, docstrings from APPS, questions from GSM8K, or prompts from DS1000. (These benchmarks are further described in Section 6.) To give an indication of the amount of data removed by decontamination, Python is the language with the highest number of matches, with 558 files removed.

| Token | Description |
|---|---|
| <\|endoftext\|> | end of text/sequence |
| <fim_prefix> | FIM prefix |
| <fim_middle> | FIM middle |
| <fim_suffix> | FIM suffix |
| <fim_pad> | FIM pad |
| <reponame> | repository name |
| <filename> | file name |
| <gh_stars> | GitHub stars |
| <issue_start> | start of GitHub issue |
| <issue_comment> | start of GitHub issue comment |
| <issue_closed> | GitHub issue closed event |
| <jupyter_start> | start of Jupyter notebook |
| <jupyter_text> | start of Jupyter text cell |
| <jupyter_code> | start of Jupyter code cell |
| <jupyter_output> | start of Jupyter output cell |
| <empty_output> | output cell without content |
| <commit_before> | code snippet before commit |
| <commit_msg> | commit message |
| <commit_after> | code snippet after commit |

Table 10: Overview of the sentinel tokens.

### 5.3 Tokenizer

The model's tokenizer follows our insights presented in Ben Allal et al. (2023) and uses those same design choices: we use the Hugging Face Tokenizers library (MOI et al., 2022) to train a byte-level Byte-Pair-Encoding with a vocabulary size of 49,152 tokens—including the sentinel tokens from table 10. The pre-tokenization step includes a digit-splitter and the regex splitter from the GPT-2 pre-tokenizer.

### 5.4 Model Architecture

We trained a 15.5B parameter model with the same architecture as SantaCoder (Ben Allal et al., 2023). It is a decoder-only Transformer with Multi-Query-Attention (MQA; Shazeer, 2019), and learned absolute positional embeddings. We also apply Fill-in-the-Middle (FIM; Bavarian et al., 2022) transformations to the training data, see Section 5.1. We used FlashAttention (Dao et al., 2022) to speed up the attention computation and reduce its memory footprint, allowing us to scale to a 8K context length. To make FlashAttention work with MQA during training, we simply expand the key and value before calling the attention kernel. The architecture hyper-parameters are given in Table 11. In addition, we have included the hyperparameters of SantaCoder(Ben Allal et al., 2023) for comparison.

### 5.5 Training details

**StarCoderBase** The model was trained for 250k iterations, with a batch size of 4M tokens, for a total of one trillion tokens. We used Adam (Kingma & Ba, 2015) with $\beta_1 = 0.9$, $\beta_2 = 0.95$, $\epsilon = 10^{-8}$ and a weight decay of 0.1. The learning rate followed a cosine decay from $3 \times 10^{-4}$ to $3 \times 10^{-5}$ after a linear warmup of 2,000 iterations.

**StarCoder** Starting from StarCoderBase, we fine-tuned a Python variant of the model for 2 epochs on the Python subset of the training data. We used the same settings as StarCoderBase, except that we used a learning rate of $5 \times 10^{-5}$ and decayed it to $5 \times 10^{-6}$ after 1,000 iterations of linear warmup. We trained for 8,500 steps.

| Hyperparameter | SantaCoder | StarCoder |
|---|---|---|
| Hidden size | 2048 | 6144 |
| Intermediate size | 8192 | 24576 |
| Max. position embeddings | 2048 | 8192 |
| Num. of attention heads | 16 | 48 |
| Num. of hidden layers | 24 | 40 |
| Attention | Multi-query | Multi-query |
| Num. of parameters | $\approx$ 1.1B | $\approx$15.5B |

Table 11: Model architecture of StarCoder. We also include SantaCoder (prior work by the community).

### 5.6 Multi-Node GPU Setup

We trained our model on a GPU cluster with 512 A100 80 GB GPUs distributed across 64 nodes. We partitioned the model with a 3D-parallel layout that shards the model with both tensor and pipeline parallelism rank 4, requiring 16 GPUs (two nodes) for one replica. To fully leverage the cluster's capabilities, we used 32-fold data parallelism. To optimize GPU utilization and reduce idle compute bubbles, we maintained a micro-batch size of 1 and accumulated for 16 steps, resulting in a global batch size of 512 (equivalent to 4M tokens). We used Megatron-LM's distributed optimizer because we found that it leads to slightly higher throughput in this configuration. Since it requires the gradient reduction step in FP32, the training in BF16 leads to 10% lower throughput than FP16, but we used it anyway to avoid training instabilities.

Except for a few restarts, we did not experience significant training instabilities.

### 5.7 CO2 emissions

**StarCoderBase** We report the carbon footprint (Lacoste et al., 2019) of training StarCoderBase. Based on the total number of GPU hours that training took (320,256) and an average power usage of 280W per GPU, this adds up to 89671.68 kWh of electricity consumed during the training process. Multiplied by the carbon intensity of the energy of the us-west-2 AWS location (0.15495 kgCO2e per kWh) and the average Power Usage Effectiveness of 1.2 across AWS datacenters, this results in 16.68 tonnes of CO2eq emitted.

**StarCoder** The fine-tuned model adds 3.5% of training time, which translates to an additional estimated emission of 0.58 tonnes of CO2eq.

## 6 Evaluation

In this section, we first outline the models we evaluated in addition to StarCoder and StarCoderBase. Then we report on the Python language performance of all models on the HumanEval (Chen et al., 2021), MBPP (Austin et al., 2021), and DS-1000 (Lai et al., 2022) evaluation benchmarks. Then we cover multi-language evaluation using a variety of benchmarks and tasks.

**A Code LM Evaluation Harness** To enable reproducible and centralized evaluation of StarCoder and other Code LLMs, we developed a Code LM Evaluation Harness (Ben Allal et al., 2022), inspired by the LM Evaluation-Harness (Gao et al., 2021b). This harness provides a framework for the efficient evaluation of code models, utilizing data parallelism and docker containers for execution. It supports several benchmarks, including HumanEval, MultiPL-E, and DS-1000.

**Other Models Evaluated** We compare StarCoder and StarCoderBase to the following models.

1. **CodeGen-16B-Multi** (Nijkamp et al., 2023) is an open-access, 16B parameter model that is trained on the Pile (Gao et al., 2021a), and then on additional code written in C, C++, Go, Java, JavaScript, and Python from the GitHub BigQuery dataset (Smith, 2016).

| Model | Size | HumanEval | MBPP |
|---|---|---|---|
| *Open-access* | | | |
| LLaMA | 7B | 10.5 | 17.7 |
| LLaMA | 13B | 15.8 | 22.0 |
| SantaCoder | 1.1B | 18.0 | 35.0 |
| CodeGen-Multi | 16B | 18.3 | 20.9 |
| LLaMA | 33B | 21.7 | 30.2 |
| CodeGeeX | 13B | 22.9 | 24.4 |
| LLaMA-65B | 65B | 23.7 | 37.7 |
| CodeGen-Mono | 16B | 29.3 | 35.3 |
| StarCoderBase | 15.5B | 30.4 | 49.0 |
| StarCoder | 15.5B | 33.6 | 52.7 |
| *Closed-access* | | | |
| LaMDA | 137B | 14.0 | 14.8 |
| PaLM | 540B | 26.2 | 36.8 |
| code-cushman-001 | 12B | 33.5 | 45.9 |
| code-davinci-002 | 175B | 45.9 | 60.3 |

Table 12: Comparing StarCoder's performance (pass@1) on the HumanEval and MBPP Python with several other models. StarCoder and StarCoder base obtain the highest performance of open-access models, and comparable performance to the code-cushman-001 closed access model.

2. **CodeGen-16B-Mono** is a version of CodeGen-16B-Multi that is fine-tuned on additional Python code from GitHub, though the dataset is not publicly available.

3. **CodeGeeX** (Zheng et al., 2023) is an open-access 13B parameter model trained on 23 programming languages selected from the Pile, the CodeParrot dataset (Wolf et al., 2020), and additional data for Python, Java, and C++. CodeGeeX also includes its own multi-language benchmark suite, HumanEval-X, which we discuss below.

4. **code-cushman-001** is a 12B parameter model by OpenAI and was the initial model for GitHub Copilot (Chen et al., 2021). The details of its training set are unknown. This model has been deprecated by OpenAI but was available from the Microsoft Azure OpenAI Service at the time of writing.[12]

5. Finally, although they are not specifically trained for code generation, we include some results from the LLaMA (Touvron et al., 2023), PaLM (Chowdhery et al., 2022), and LaMDA (Thoppilan et al., 2022) papers. LLaMA's license prohibits commercial use, and PaLM and LaMDA are not publicly available.

## 6.1 StarCoder: Python Evaluation

In this section, we evaluate the performance of StarCoder on Python, comparing it to both open-access and closed-access models. We first report performance on HumanEval (Chen et al., 2021) and MBPP (Austin et al., 2021), which are two widely used benchmarks of Python performance. However, we also measure performance on DS-1000 (Lai et al., 2022), a code completion benchmark of 1,000 Python data science problems based on StackOverflow questions.

### 6.1.1 The HumanEval and MBPP Benchmarks

HumanEval (Chen et al., 2021), and MBPP (Austin et al., 2021) are widely-used benchmarks for Code LLMs consisting of hundreds of Python programming problems that use test cases to validate the code produced by

---

[12] There had been a code-cushman-002, but it is not available at the time of writing.

| Format | Model | Matplotlib | NumPy | Pandas | PyTorch | SciPy | Scikit-Learn | TensorFlow | Overall |
|---|---|---|---|---|---|---|---|---|---|
| | Number of problems: | 155 | 220 | 291 | 68 | 106 | 115 | 45 | 1,000 |
| Completion | SantaCoder-1B | 21.6 | 4.6 | 0.9 | 2.6 | 2.4 | 4.8 | 3.1 | 5.7 |
| Completion | InCoder-6B | 28.3 | 4.4 | 3.1 | 4.4 | 2.8 | 2.8 | 3.8 | 7.4 |
| Completion | CodeGen-16B-Mono | 31.7 | 10.9 | 3.4 | 7.0 | 9.0 | 10.8 | 15.2 | 11.7 |
| Completion | code-cushman-001 | 40.7 | 21.8 | 7.9 | 12.4 | 11.3 | 18.0 | 12.2 | 18.1 |
| Completion | StarCoderBase | 47.0 | 27.1 | 10.1 | 19.5 | **21.7** | 27.0 | 20.5 | 23.8 |
| Completion | StarCoder | **51.7** | **29.7** | **11.4** | **21.4** | 20.2 | **29.5** | **24.5** | **26.0** |
| Insertion | SantaCoder-1B | 21.6* | 13.8 | 2.0 | 3.8 | 5.7 | 6.9 | 14.8 | 9.3 |
| Insertion | InCoder-6B | 28.3* | 4.6 | 2.9 | 4.4 | 2.8 | 3.1 | 7.8 | 7.5 |
| Insertion | StarCoderBase | 47.0* | 26.3 | **10.9** | 16.6 | **20.2** | **30.2** | **22.3** | 24.0 |
| Insertion | StarCoder | **51.7*** | **30.8** | 10.3 | **21.0** | **20.2** | 27.4 | 20.0 | **25.4** |

Table 13: Performance of open-access and closed-access models on DS-1000. Benchmarks are as follows. All models evaluated at temperature=0.2, top_p=0.5, max_length=1024. Scores reflect mean pass@1 accuracy averaged over 40 samples. *: Matplotlib task does not have right sided context, so insertion and completion formats are identical.

a Code LLM. Code LLMs generate code by sampling from their output distribution. We report performance using the pass@$k$ metric (Chen et al., 2021): the total fraction of benchmark problems solved, where a problem is considered solved if any one of $k$ code samples passes every test case. Like Chen et al. (2021), we use sampling temperature 0.2 for pass@1, and temperature 0.8 for $k > 1$. We generate $n = 200$ samples for all experiments with open-access models. For API models, we use $n = 20$ samples, which is enough to estimate pass@1. We focus on the simplest version of pass@$k$, which is pass@1: the likelihood that a problem is solved in a single attempt by the model.

Table 12 compares StarCoder (and StarCoderBase) on HumanEval and MBPP to several open-access and closed-access models:

1. *StarCoder is the highest-performing open-access model on both benchmarks.*

2. *StarCoder outperforms the largest models*, including PaLM, LaMDA, and LLaMA, despite being significantly smaller.

3. *StarCoderBase is also very capable on Python* and is competitive with CodeGen-16B-Mono, a similarly-sized open-access model that was fine-tuned on Python.

4. *StarCoder outperforms OpenAI's code-cushman-001 (12B) model.*

### 6.1.2 The DS-1000 Python Data Science Benchmarks

A major limitation of HumanEval and MBPP is that they are simple programming puzzles that are not representative of the code that most programmers write. In contrast, the DS-1000 benchmark (Lai et al., 2022) has a suite of 1,000 realistic and practical data science workflows across seven libraries and evaluates generations in execution against test cases.

DS-1000 supports two evaluation modes: completion and insertion (via FIM). We report completion scores for all models but insertion scores only for models that support it: the StarCoder models and InCoder-6B (Fried et al., 2022). DS-1000 also categorizes problems based on the libraries used: Matplotlib, NumPy, Pandas, SciPy, Scikit-Learn, PyTorch, and TensorFlow. We report pass@1 for each library and an overall score in Table 13 and draw the following conclusions:

1. *StarCoder substantially outperforms all other models on data science problems* from the DS-1000 benchmark. Moreover, this is true across every kind of data science library.

2. *StarCoderBase also outperforms every other model*, but is slightly behind StarCoder on DS-1000.

3. We confirm the finding by Lai et al. (2022): *model performance on HumanEval and MBPP benchmarks does not always correlate with performance on the more realistic DS-1000 benchmarks.* For example, CodeGen-Mono slightly outperforms code-cushman-001 and the StarCoder models on HumanEval and MBPP, but is significantly worse on DS-1000. This demonstrates the importance of evaluating models on a range of benchmarks.

### 6.1.3 The ODEX Open-Domain Coding Benchmark

Our previous evaluations focus either on *closed domains* (i.e., primarily built-in Python functions, as in MBPP and HumanEval) or specific domains (e.g., data science, as in DS-1000). To evaluate model ability to generate code on a broader set of Python libraries, we use the ODEX benchmark (Wang et al., 2022) containing 505 open-domain and 440 closed-domain Python coding queries, in four natural languages — English, Spanish, Japanese, and Russian — with test-case-based execution evaluation.

We report the pass@1 metric for StarCoder and baseline models, including Codex (code-davinci-001), CodeGen-16B-Mono, and SantaCoder. In addition to the overall execution accuracy, we also categorize problems by languages and domains, which are: (1) queries in the *closed-domain* (using only built-in Python functions) and *open-domain* (using functions from imported libraries), and (2) queries with instructions written in English, Spanish, Japanese, and Russian, respectively. We report overall scores and scores in different domains and languages in Table 14 and draw the following conclusions:

1. *StarCoder substantially outperforms all other models on open-domain coding queries* from the ODEX benchmark.

2. *StarCoderBase also outperforms every other model*, even better than StarCoder in the ODEX English subset, but slightly behind in other languages.

3. Both StarCoder and StarCoderBase models generally exhibit smaller gaps between open- and closed-domain queries than other baseline models, despite the higher overall execution accuracy. This result indicates that StarCoder models acquire more generalized skills about coding queries in the open domain (i.e., concerning diverse Python libraries), while other models exhibit larger performance drops when moving from the closed to open domain.

| Model | English | | | Spanish | | | Japanese | | | Russian | | |
|---|---|---|---|---|---|---|---|---|---|---|---|---|
| | overall | open | closed | overall | open | closed | overall | open | closed | overall | open | closed |
| CodeGen-16B-Mono | 33.7 | 25.2 | 43.1 | 30.0 | 25.0 | **43.1** | 37.8 | 26.6 | **62.8** | 46.8 | 30.4 | 60.1 |
| code-cushman-001 | 31.9 | 24.4 | 40.2 | 31.9 | 27.7 | 36.7 | 25.7 | 21.2 | 35.5 | 40.0 | 26.0 | 51.6 |
| code-davinci-001 | 33.6 | 26.9 | 41.0 | 36.9 | 31.7 | 42.9 | 31.0 | 23.7 | 47.3 | 43.2 | 28.9 | 55.1 |
| SantaCoder | 37.7 | 30.9 | 45.1 | 32.1 | 26.0 | 39.1 | 28.1 | 23.0 | 39.4 | 36.9 | 23.0 | 48.3 |
| StarCoderBase | **46.5** | **40.7** | 53.0 | 30.1 | 25.4 | 35.5 | 41.2 | 37.6 | 49.2 | 46.1 | 34.0 | 56.1 |
| StarCoder | 44.7 | 37.0 | **53.1** | **37.6** | **32.9** | 42.9 | **44.2** | **39.6** | 54.5 | **50.4** | **33.8** | **64.1** |

Table 14: Performance on the ODEX benchmark by instruction languages and code domains: *open* problems use libraries, while *closed* use only built-in Python functions.

## 6.2 StarCoder and StarCoderBase: Multi-Language Evaluation

In this section, we focus primarily on StarCoderBase, and evaluate its performance on a variety of programming languages and programming tasks, including producing code from natural language descriptions, documenting code, predicting type annotations, and more. This section also shows that StarCoder, despite being fine-tuned on Python, remains a very capable multi-language Code LLM and even outperforms StarCoderBase on some languages.

| Language | CodeGen-16B-Multi | CodeGeeX | code-cushman-001 | StarCoder | StarCoderBase |
|---|---|---|---|---|---|
| cpp | 21.00 | 16.87 | 30.59 | **31.55** | 30.56 |
| c-sharp | 8.24 | 8.49 | **22.06** | 21.01 | 20.56 |
| d | 7.68 | 9.15 | 6.73 | **13.57** | 10.01 |
| go | 13.54 | 11.04 | 19.68 | 17.61 | **21.47** |
| java | 22.20 | 19.14 | **31.90** | 30.22 | 28.53 |
| julia | 0.00 | 0.29 | 1.54 | **23.02** | 21.09 |
| javascript | 19.15 | 16.92 | 31.27 | 30.79 | **31.70** |
| lua | 8.50 | 10.96 | 26.24 | 23.89 | **26.61** |
| php | 8.37 | 13.51 | **28.94** | 26.08 | 26.75 |
| perl | 3.42 | 8.09 | **19.29** | 17.34 | 16.32 |
| python | 19.26 | 21.62 | 30.71 | **33.57** | 30.35 |
| r | 6.45 | 3.92 | 10.99 | **15.50** | 10.18 |
| ruby | 0.00 | 3.34 | **28.63** | 1.24 | 17.25 |
| racket | 0.66 | 3.31 | 7.05 | 0.07 | **11.77** |
| rust | 4.21 | 7.88 | **25.22** | 21.84 | 24.46 |
| scala | 2.37 | 8.95 | 27.62 | 27.61 | **28.79** |
| bash | 0.61 | 2.75 | **11.74** | 10.46 | 11.02 |
| swift | 1.25 | 7.26 | 22.12 | **22.74** | 16.74 |
| typescript | 20.07 | 10.11 | 31.26 | **32.29** | 32.15 |

Table 15: Comparing StarCoder to multi-language open-access (e.g., CodeGen-16B-Multi) and closed-access models (e.g., code-cushman-001) on 19 programming languages. We report pass@1 on HumanEval (Chen et al., 2021), which we translate from Python to the other languages using MultiPL-E (Cassano et al., 2023).

### 6.2.1 Evaluation on 19 Programming Languages with MultiPL-E

We evaluate the ability of StarCoder to turn natural language into working code in multiple programming languages using MultiPL-E (Cassano et al., 2023), which translates the HumanEval (Chen et al., 2021) and MBPP (Austin et al., 2021) Python benchmarks into 18 other programming languages as follows.

MultiPL-E has a set of rule-based compilers that translate Python benchmarks to each target programming language. Each compiler expects a benchmark in the HumanEval format: 1) a natural language description (in a docstring), 2) a function signature (name, arguments, and, potentially, types), and 3) a set of hidden assertions. The MultiPL-E compilers translate the function signature, assertions, and docstring (which may have doctests) into a target language. Thus, MultiPL-E gives us a parallel set of benchmarks derived from HumanEval and MBPP to compare model performance across programming languages.[13] The MultiPL-E languages include both high and low-resource languages, statically and dynamically typed languages, and a variety of other programming language features.

Table 15 shows how these models perform on 19 programming languages, and from it, we draw the following conclusions:

1. Across all 19 programming languages, *StarCoderBase outperforms other open-access models, sometimes showing more than 2× performance.*

2. *StarCoderBase is competitive with code-cushman-001 on most languages that we evaluate.* There are a few exceptions. For example, code-cushman-001 outperforms StarCoderBase by more than 5% on C++, Java, Ruby, and Swift, and StarCoder outperforms code-cushman-001 by more than 5% on Julia.

---

[13]The MultiPL-E prompts are slightly different from the original HumanEval and MBPP prompts. For example, in HumanEval, some ad hoc examples in docstrings are reformatted to be doctests so that they can be translated into examples in each target language. MultiPL-E also omits three HumanEval benchmarks that do not fit the above format. These changes have a small impact on pass rates.

| Format | Model | Valid (↑) | Insecure (↓) |
|--------|-------|-----------|--------------|
| Completion | StarCoderBase | 855/1000 (85.50%) | 340/855 (39.77%) |
| Insertion | StarCoderBase | **987/1000 (98.70%)** | 354/987 (35.87%) |
| Completion | InCoder-6B | 871/1000 (87.10%) | 309/871 (35.48%) |
| Insertion | InCoder-6B | 854/1000 (85.40%) | **293/854 (34.31%)** |
| Completion | CodeGen-16B-Multi | 955/1000 (95.50%) | 413/955 (43.25%) |
| Completion | code-cushman-001 | 964/1000 (96.40%) | 408/964 (42.32%) |

Table 16: Performance on the *Asleep at the Keyboard* security benchmark (Pearce et al., 2022).

3. *Despite fine-tuning on Python, StarCoder remains competitive on most languages*, and also outperforms other open models. What is more surprising is that *StarCoder slightly outperforms StarCoderBase on certain languages*, despite being fine-tuned on Python. At this time, we can only speculate on why this is the case, and further investigation of the open training data is likely to help shed light on this finding.

There are several other conclusions that we can draw from the table. For example, CodeGen-16B-Multi performs better than one might expect on some languages that are reportedly not in its training set, including C#, Lua, PHP, and TypeScript. Its performance on TypeScript is less surprising since simple JavaScript functions often type-check with TypeScript by design. Similarly, StarCoder shows high performance on Swift, even though it was not included in its training set, as explained in Section 3.1.

### 6.2.2 The "Asleep at the Keyboard" Security Benchmark

A limitation of Code LLMs is that they can generate code with security vulnerabilities (Pearce et al., 2022). The *Asleep at the Keyboard* benchmark by Pearce et al. (2022) has 89 security-sensitive scenarios across three evaluation axes: (1) Diversity of Weakness (DoW) covers 18 different vulnerability classes in MITRE's Common Weakness Enumeration (CWE) taxonomy, with scenarios drawn from the 2021 CWE Top 25 Most Dangerous Software Weaknesses list published by MITRE; (2) Diversity of Prompt (DoP) evaluates the model's sensitivity to variations in the prompt for a single vulnerability class (SQL injection); (3) Diversity of Domain (DoD) contains security scenarios in the hardware description language Verilog. We focus on the DoW, which contains 54 scenarios (25 in C and 29 in Python) across 18 CWEs. We exclude scenarios that lack an automated test, leaving 40 scenarios (23 in C and 17 in Python).

Pearce et al. (2022) had previously evaluated the security of GitHub Copilot (as of August 2021), and in this paper, we use the same methodology to evaluate StarCoderBase, InCoder-6B, CodeGen-16B-Multi, and OpenAI's code-cushman-001. We use the original benchmarking methodology: generating 25 completions per scenario at temperature 0.2 (1,000 completions per model). The dataset supports fill-in-the-middle, so we include this configuration on models that support it. The results are shown in Table 16; **Valid** gives the percentage of solutions that were syntactically valid (using `py_compile` for Python and `gcc` for C), and **Insecure** shows the percentage of *valid* solutions that contained the vulnerability the scenario tests for. From this table, we draw the following conclusions.

1. *StarCoderBase has the highest rate of valid code.*

2. *InCoder-6B has a slightly lower rate for insecure code generation, but this may be due to its lower rate of valid completions.*

3. Among the models with more than 95% valid code, StarCoder has the lowest rate of insecure completions.

### 6.2.3 Fill in the Middle Benchmarks

The StarCoder models support *fill in the middle* (FIM) or *infilling*, which allows the model to generate code conditioned on prefix and suffix code surrounding the insertion point. Only a handful of recent models

| Model | Java | JavaScript | Python |
|---|---|---|---|
| InCoder-6B | 0.49 | 0.51 | 0.31 |
| SantaCoder | 0.62 | 0.60 | 0.44 |
| StarCoder | **0.73** | **0.74** | **0.62** |

Table 17: Performance on single-line fill-in-the-middle on the FIM benchmark by Ben Allal et al. (2023).

| Model | Non-None F1 | All F1 |
|---|---|---|
| InCoder-6B | 59.1 | 46.8 |
| SantaCoder | 66.9 | 78.5 |
| StarCoderBase | **77.4** | **86.6** |
| StarCoder | 77.1 | 86.4 |

Table 18: Accuracy of Python return type prediction, using Fried et al. (2022)'s adaptation of the Pradel et al. (2020) benchmarks. We report both the overall F1 scores, which include trivial None-type prediction, and the F1 score for non-None types.

support FIM: from OpenAI (Bavarian et al., 2022), InCoder (Fried et al., 2022), and our prior work on SantaCoder (Ben Allal et al., 2023). FIM opens up the possibility of a variety of tasks that go beyond left-to-right code completion. We evaluate StarCoderBase on four established FIM benchmarks below.

**Single-Line Infilling for Python, Java, and JavaScript**  Fried et al. (2022) present a single-line fill-in-the-middle task for Python that masks one line of code from a HumanEval solution and scores the model's ability to complete the function. They turn every HumanEval solution into several fill-in-the-middle problems by masking each non-blank, non-comment line of code in the solution body into a fill-in-the-middle task. Ben Allal et al. (2023) generalizes this benchmark to also support Java and JavaScript, using model-generated solutions from MultiPL-E's translations. We compare the performance of StarCoderBase, SantaCoder, and InCoder on this task, evaluating using line exact match (Table 17). StarCoderBase significantly outperforms the two smaller models.

**Python Return Type Prediction**  Pradel et al. (2020) introduce methods and datasets for evaluating Python type annotations. Fried et al. (2022) adapt and filter one dataset from this work, consisting of Python functions from GitHub, and use it to evaluate infilling models on function return type prediction. We use this dataset to compare StarCoder, StarCoderBase, and SantaCoder to InCoder on function return type prediction. Our setup follows Fried et al. (2022): each model uses greedy generation to infill return types while conditioning on the imports, body, and signature for each function. We report exact match accuracy on normalized annotations for all functions in the evaluation set and only those with non-None annotations, following Fried et al. (2022). We find that *StarCoder and StarCoderBase outperform existing approaches at Python return type prediction* (Table 18). However, we note that as the functions in this evaluation set were taken from GitHub repositories, they may overlap with the training data for SantaCoder and the StarCoder models.

**TypeScript Type Prediction**  Yee & Guha (2023) evaluate approaches to neural type prediction for TypeScript. However, instead of measuring accuracy, they argue that benchmarks should measure how many projects or files do not have type errors with predicted types. This approach makes it possible to evaluate type prediction for JavaScript programs that have never been translated to TypeScript, which reduces the likelihood of dataset contamination. We add StarCoderBase to their evaluation framework and compare it to InCoder, which performs best at type prediction in the original work. Table 19 shows that StarCoderBase outperforms InCoder: (1) it produces more packages that type check, (2) across all packages, it produces more files that type check, and (3) it produces fewer trivial type annotations than InCoder.

| | Packages type check | | | Files with no errors | | | Trivial annotations | | |
|---|---|---|---|---|---|---|---|---|---|
| | ✓ | Total | % | ✓ | Total | % | ✓ | Total | % |
| InCoder | 30 | 128 | 23.4 | 571 | 760 | 75.1 | 56 | 117 | 47.9 |
| StarCoderBase | 49 | 128 | 38.3 | 593 | 760 | 78.0 | 135 | 299 | 45.2 |

Table 19: TypeScript type prediction performance using the dataset and metholody from Yee & Guha (2023). We only evaluate JavaScript packages that have never been translated to TypeScript and compare StarCoder to InCoder, the best-performing model by Yee & Guha (2023). StarCoder outperforms InCoder in several ways.

| Model | BLEU |
|---|---|
| InCoder-6B | 18.27 |
| SantaCoder | 19.74 |
| StarCoderBase | 21.38 |
| StarCoder | **21.99** |

Table 20: Performance on the Python portion of the CodeXGLUE Code Summarization task, evaluating function docstring generation. Models are evaluated zero-shot using their infilling capability.

**Python Docstring Generation**  To evaluate models' ability to generate documentation for functions, we use the Python subset of the CodeXGLUE code summarization benchmark (Lu et al., 2021). This benchmark is constructed from the CodeSearchNet dataset (Husain et al., 2019), containing functions from public GitHub repositories. Models infill the documentation string (docstring) for each function using greedy decoding, conditioned on the function signature and body. We follow the evaluation scheme of past work: docstrings are evaluated using smoothed 4-gram BLEU (Papineni et al., 2002) against the reference docstring from the original function, using only the first lines of the generated and reference docstrings (removing, e.g., descriptions of function arguments and return types that may appear in later lines). In Table 20, we see that *StarCoder and StarCoderBase obtain higher performance than past work on docstring generation.* However, we note that there may be an overlap between this evaluation dataset and the data used to train SantaCoder and the StarCoder models.

### 6.3   Performance Improvement Through the Training Process

We evaluate the performance of StarCoderBase at several training checkpoints after every 200B tokens seen out of the total 1000B. Figure 2 (right) shows how performance (pass@1) changes during training for each programming language supported by MultiPL-E. The performance curve for several high-resource programming languages suggests that training longer is likely to improve their performance further.

However, some of the low-resource languages see limited improvement during training or even have a pass@1 decline. For example, R's pass@1 rate drops significantly between the 800B and 1000B (final) checkpoints. The dependence of pass@1 on data size (Figure 2, left) further supports the hypothesis that this is related to the amount of data available. The slope of the linear fit increases between 800B and 1000B checkpoints while the intercept decreases, i.e., performance improves only for languages with large enough amounts of data ($\gtrsim 1$ GB).

We manually inspected the completions generated by R over several checkpoints to better understand model performance. One might hypothesize that some problems are harder than others, and so the model gains and loses the ability to solve them in R over the 600B, 800B, and 1000B checkpoints, but we find that this is *not* the case. Instead, we find significant variance in per-problem success rates for several problems (Table D.3). For these problems, the pass rate between different checkpoints varies in what appears to be a completely uncorrelated manner. Moreover, manual inspection shows that the failures are caused by minor mistakes,

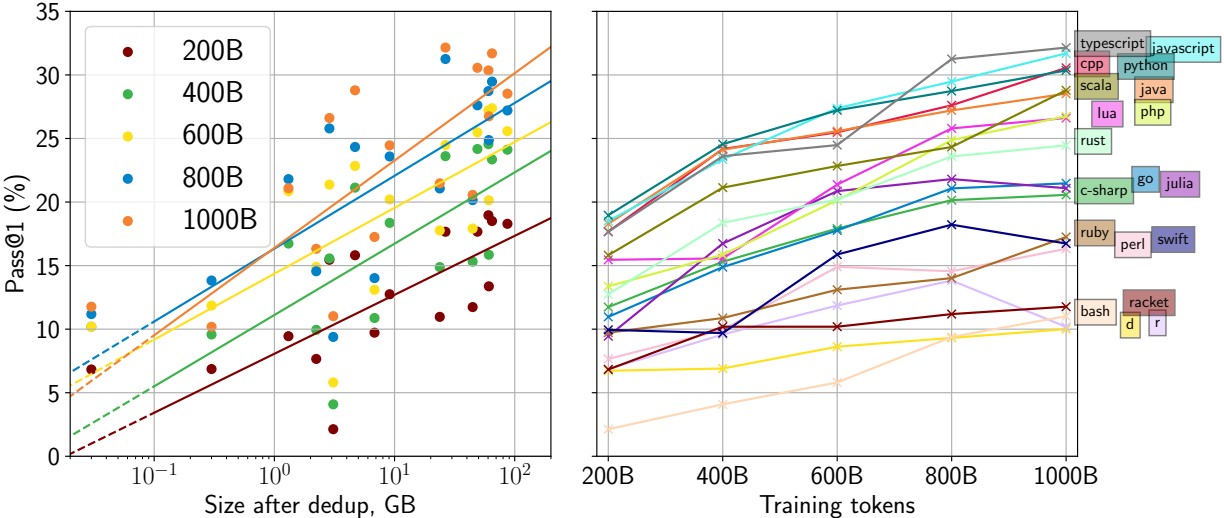

Figure 2: Performance (pass@1) of StarCoderBase at several training checkpoints by data size **(left)** and by programming language **(right)**. The lines in the left plot are a linear fit between pass@1 and log-dataset-size for all the points except the leftmost one, where we expect the linear dependence to break due to transfer learning (dashed line). The goodness of fit ranges between $R^2 = 0.399$ for the 600B checkpoint to $R^2 = 0.510$ for the 1000B checkpoint.

e.g., not taking the absolute value when computing GCD, not converting a string to a character array, or not checking edge cases.

### 6.4 Perplexity With Long Contexts

StarCoderBase was trained with an 8K token window, allowing conditioning on and generating long code files. To evaluate the ability of the model to benefit from this larger context, we compare its perplexity (Bahl et al., 1983) when using a full window size of 8K tokens versus a window size of 2K tokens (as used in many prior code models).

To ensure no overlap between the training data for StarCoderBase and the perplexity computation data, we downloaded 10 GNU Public License (GPL) repositories from GitHub in each of the languages in Table 21. We compiled all files from the repositories into a single document for each language. We then divided these documents into 8K token chunks and computed perplexity on the last 1K tokens in each chunk[14] in two conditions: (1) the model window only contains the final 2K tokens in the chunk (i.e., the 1K being predicted and the previous 1K), and (2) the model window contains all 8K tokens in the chunk (i.e., the 1K tokens being predicted and the previous 7K). This evaluates the ability of the model to benefit from additional file- and repo-level context when predicting code. In Table 21, we report the average perplexity of the 1K token regions across all chunks. We see that StarCoderBase indeed benefits from the extra token conditioning afforded by its 8K context window, with substantially lower perplexities across all languages.

## 7 Natural Language Evaluation

Although the StarCoder models are principally developed to be Code LLMs, they have also been trained on a significant amount of natural language text. Roughly 20% of its training tokens are natural language data: 7% GitHub issues, 10% Markdown, 2% Jupyter notebooks, and 4% HTML. In this section, we evaluate

---

[14]We evaluate perplexity on the final 1K tokens in each 8K chunk so that both conditions have the same evaluation tokens, and to avoid overly penalizing the 2K condition, as tokens at the beginning of a window tend to have higher perplexity as there is less context available to predict them.

| Window Size | Language | | | | | | | | | |
|---|---|---|---|---|---|---|---|---|---|---|
| | cpp | c-sharp | c | go | java | javascript | php | r | ruby | rust |
| 2K tokens | 2.01 | 1.90 | 1.71 | 1.35 | 1.65 | 1.98 | 1.73 | 1.72 | 2.16 | 1.84 |
| 8K tokens | **1.79** | **1.66** | **1.61** | **1.21** | **1.54** | **1.68** | **1.43** | **1.48** | **2.02** | **1.65** |

Table 21: Perplexity of StarCoderBase on evaluation regions (of size 1K tokens) when using a window size of 2K or 8K tokens across repositories from 10 languages. The larger window size substantially reduces perplexity, demonstrating a benefit of StarCoder's 8K token window.

| Model | Size | GSM8K CoT | +maj1@100 | GSM8K PAL | +maj1@40 |
|---|---|---|---|---|---|
| StarCoderBase | 15.5B | 8.4 | — | 21.5 | 31.2 |
| CodeGen-Multi | 16B | 3.18 | — | 8.6 | 15.2 |
| CodeGen-Mono | 16B | 2.6 | — | 13.1 | 22.4 |
| | 7B | 11.0 | 18.1 | 10.5 | 16.8 |
| | 13B | 17.8 | 29.3 | 16.9 | 28.5 |
| LLaMA | 33B | 35.6 | 53.1 | 38.7 | 50.3 |
| | 65B | **50.9** | **69.7** | — | — |

Table 22: 8-shot accuracy on the GSM8K math-reasoning benchmark. Samples are generated with greedy decoding. `maj1@k` denotes a majority vote over $k$ generations. For the majority vote, we instead generate samples using nucleus sampling with $p = 0.95$ and temperature 0.7, following Gao et al. (2022). We use "—" when a model was not evaluated on a given metric, or the metric is not supported in Language Model Evaluation Harness. The LLaMA CoT numbers are from Touvron et al. (2023).

StarCoderBase on several natural language tasks: natural language reasoning and understanding tasks that might benefit from the combination of code and text training data; and natural language generation tasks that evaluate the model's tendencies to produce undesirable text outputs, e.g., in a documentation generation or interactive assistant setting.

## 7.1 Math Reasoning

Recent work has shown that Code LLMs can be effective arithmetic and symbolic reasoners by using a technique called Program-Aided Language models (PAL; Gao et al., 2022). With PAL, the LLM reads the reasoning problem and generates Python programs as the intermediate reasoning steps, which are then executed by the Python interpreter to produce the answer. In contrast, the Chain-of-Thought method (CoT; Wei et al., 2022) prompts the LLM to produce the reasoning steps in natural language before generating the answer.

We investigate the reasoning capabilities of StarCoderBase on GSM8K (Cobbe et al., 2021), a set of middle-school math word problems. We compare with the two CodeGen-16B models (Nijkamp et al., 2023) and the family of LLaMA models (Touvron et al., 2023). The results of our evaluation are presented in Table 22, where we provide both CoT and PAL results for StarCoderBase and LLaMA.

In line with previous results comparing PAL to CoT on Code LLMs (Gao et al., 2022), we find that StarCoderBase performs better with PAL (21.5%) than with CoT (8.4%). StarCoderBase substantially outperforms CodeGen-16B-Mono and CodeGen-16B-Multi, which achieve 13.1% and 8.6% with PAL, respectively. These differences carry over to the setting where majority voting is applied. The difference between CoT and PAL is much smaller for the LLaMA models, although we observe that CoT performs slightly better for the 7B and 13B LLaMA models. Interestingly, we find that StarCoderBase outperforms LLaMA-13B (17.8%) on this reasoning benchmark. However, its performance still lags behind LLaMA-33B (38.7%).

| Model | Size | MMLU 5-shot acc, % |
|---|---|---|
| CodeGen-Multi | 16B | 27.8 |
| GPT-NeoX | 20B | 32.9 |
| StarCoder | 15.5B | 33.9 |
| StarCoderBase | 15.5B | 34.2 |
| LLaMA | 7B | 35.1 |
| LLaMA | 13B | **46.9** |

Table 23: 5-shot accuracy on the MMLU language understanding benchmark.

| Model | Size | CoQA zero-shot F1 score |
|---|---|---|
| CodeGen-Multi | 16B | 0.59 |
| StarCoderBase | 15.5B | 0.67 |
| StarCoder | 15.5B | 0.67 |
| LLaMA | 7B | 0.71 |
| LLaMA | 13B | **0.73** |
| GPT-NeoX | 20B | **0.73** |

Table 24: Zero-shot accuracy on the CoQA question answering challenge.

## 7.2 World Knowledge and Reading Comprehension

MMLU (Hendrycks et al., 2020) is a massive multitask language understanding benchmark, covering multiple-choice questions in 57 knowledge domains, including the humanities, STEM, and social sciences. CoQA (Reddy et al., 2019) is a large-scale dataset for Conversational Question Answering systems, measuring the model's ability to process a text passage and answer a series of interconnected questions. We compare StarCoderBase and StarCoder with CodeGen-16B-Multi (Nijkamp et al., 2023), GPT-NeoX (Black et al., 2022), LLaMA-7B, and LLaMA-13B (Touvron et al., 2023).

We present the 5-shot accuracy for MMLU in Table 23, and the zero-shot F1 scores for CoQA in Table 24. On MMLU, StarCoderBase outperforms CodeGen-16B-Multi significantly (34.2% to 27.8%), and even outperforms GPT-NeoX by a small margin (32.9%). Nevertheless, both LLaMA models outperform StarCoderBase. On CoQA, StarCoderBase performs better than CodeGen-16B-Multi but is outperformed by LLaMA and GPT-NeoX.

## 7.3 Measuring Harmful Generation

When generating open-ended text such as code documentation or technical dialogue, a Code LLM (similarly to text-only LLMs) might produce harmful outputs. We compare StarCoderBase to previous Code LLMs on benchmarks that measure social bias and toxicity in model-produced text.[15]

### 7.3.1 Social Bias

Recent work has highlighted that LLMs often capture social biases and stereotypes from their pre-training corpora (Kurita et al., 2019; May et al., 2019; Hutchinson et al., 2020; Meade et al., 2023). To quantify social bias within our model, we use StereoSet (Nadeem et al., 2021).

StereoSet consists of a collection of fill-in-the-blank-style tests for measuring social biases within language models.[16] Each example in StereoSet consists of an incomplete sentence (e.g., *our housekeeper is* BLANK)

---

[15]Code for the evaluations is available here: https://github.com/McGill-NLP/StarCoderSafetyEval
[16]We only evaluate against the intrasentence task in this work.

| Model | Stereotype Score | Language Model Score | ICAT Score |
|---|---|---|---|
| *Gender* | | | |
| LLaMA-13B | 66.54 | **88.09** | 58.95 |
| CodeGen-Multi-16B | 67.34 | 86.41 | 56.44 |
| StarCoderBase | **58.76** | 86.82 | **71.60** |
| *Profession* | | | |
| LLaMA-13B | 60.95 | **86.74** | 67.74 |
| CodeGen-Multi-16B | 60.67 | 85.67 | 67.38 |
| StarCoderBase | **53.24** | 84.70 | **79.21** |
| *Race* | | | |
| LLaMA-13B | 64.94 | 87.97 | 61.68 |
| CodeGen-Multi-16B | 60.58 | **88.60** | 69.85 |
| StarCoderBase | **56.48** | 86.82 | **75.58** |
| *Religion* | | | |
| LLaMA-13B | 57.95 | 90.26 | 75.91 |
| CodeGen-Multi-16B | 56.16 | 88.91 | 77.96 |
| StarCoderBase | **55.69** | **90.67** | **80.36** |
| *Overall* | | | |
| LLaMA-13B | 63.40 | **87.62** | 64.14 |
| CodeGen-Multi-16B | 61.29 | 87.25 | 67.55 |
| StarCoderBase | **55.53** | 86.18 | **76.65** |

Table 25: StereoSet intrasentence results for gender, professional, racial, and religious bias. Stereotype scores close to 50% are best. Language modeling scores and ICAT scores close to 100% are best.

alongside three possible completions. Of these completions, one is stereotypical (e.g., *Mexican*), another is anti-stereotypical (e.g., *Italian*) and a third is unrelated (e.g., *computer*). StereoSet defines three metrics: a stereotype score, a language modeling score, and an ICAT score. The stereotype score is the percentage of examples for which a model *prefers* the stereotypical completion for a sentence over the anti-stereotypical completion. The language modeling score is the percentage of examples for which a model prefers a meaningful completion (stereotype or anti-stereotype) over an unrelated completion. Finally, Nadeem et al. (2021) define an idealized context association test (ICAT) score that combines these two metrics:

$$\text{ICAT} = \text{lms} \cdot \frac{\min(\text{ss}, 100 - \text{ss})}{50} \tag{1}$$

where lms and ss denote the language model score and stereotype score, respectively.

We report StereoSet results for StarCoderBase, alongside LLaMA-13B and CodeGen-Multi-16B, in Table 25. Across all four bias domains, we find StarCoderBase obtains the lowest stereotype scores, but also has competitive language modeling scores. This suggests that StarCoderBase's lower stereotype scores are not simply due to worse language modeling (Meade et al., 2022), and also as indicated by the high ICAT score.

We also evaluate StarCoderBase against Crowdsourced Stereotype Pairs (CrowS-Pairs; Nangia et al. 2020) and refer readers to Table D.4 for results.

### 7.3.2 Toxicity

To evaluate toxicity in responses generated from our model, we use RealToxicityPrompts (Gehman et al., 2020), a collection of sentence-level prompts that often elicit undesirable responses from language models. We generate responses to 10K examples from RealToxicityPrompts using StarCoderBase with a minimum

| Model | Classifier | Word List |
|---|---|---|
| LLaMA-13B | 0.74 | 1.43 |
| CodeGen-Multi-16B | **0.21** | **0.82** |
| StarCoderBase | 0.42 | 1.12 |

Table 26: RealToxicityPrompts response toxicity results. We report the percentage of responses flagged as toxic using a toxicity classifier and an offensive word list. Lower scores are indicative of less toxic generations.

| Model | Size | Open Access | Synth. Reason. (AS) | Synth. Reason. (NL) | bAbI | Dyck | GSM8K | MATH | MATH (CoT) | LSAT | Legal Support |
|---|---|---|---|---|---|---|---|---|---|---|---|
| code-davinci-002 | 175B | | **54.0** | 68.4 | **68.6** | 80.5 | **56.8** | **41.0** | 43.3 | — | — |
| text-davinci-003 | 175B | | 50.2 | **73.4** | 65.3 | 75.1 | 50.6 | 39.0 | **44.9** | **23.3** | **62.2** |
| Luminous Supreme | 70B | | 31.2 | — | 50.4 | 72.9 | 11.2 | 14.9 | 5.7 | 21.2 | 53.0 |
| StarCoderBase | 15.5B | ✓ | 44.0 | 21.0 | 50.4 | **85.4** | 8.4 | 15.1 | 7.0 | 19.0 | 53.2 |
| Cohere Command Beta | 52.4B | | 24.3 | 24.5 | 47.3 | 42.1 | 13.8 | 13.3 | 7.5 | 22.9 | 60.6 |
| J1-Jumbo v1 | 178B | | 26.3 | 17.4 | 54.3 | 44.5 | 5.4 | 8.9 | 3.3 | 23.2 | 48.4 |
| J1-Grande v2 beta | 17B | | 28.6 | 13.9 | 47.0 | 61.7 | 9.6 | 12.7 | 6.8 | 19.1 | 56.2 |
| code-cushman-001 | 12B | | 34.1 | 16.4 | 48.1 | 45.1 | 4.9 | 9.9 | 7.2 | — | — |
| OPT | 175B | ✓ | 22.5 | 24.8 | 50.7 | 49.4 | 4.0 | 6.5 | 2.6 | 22.0 | 53.2 |
| GPT-NeoX | 20B | ✓ | 20.4 | 16.7 | 46.8 | 74.7 | 5.3 | 14.1 | 7.1 | 19.1 | 51.5 |
| BLOOM | 176B | ✓ | 30.4 | 19.7 | 44.7 | 54.5 | 9.5 | 4.3 | 5.5 | 20.9 | 54.3 |
| GLM | 130B | ✓ | 25.2 | 25.4 | 44.3 | 54.9 | 6.1 | 0 | 5.9 | 19.3 | 45.1 |
| UL2 | 20B | ✓ | 20.5 | 21.7 | 50.1 | 14.0 | 2.4 | 0 | 0 | 20.7 | 50.6 |
| OPT | 66B | ✓ | 19.3 | 21.3 | 40.8 | 47.1 | 1.8 | 4.8 | 2.9 | 17.5 | 52.7 |
| YaLM | 100B | ✓ | 5.6 | 6.1 | 34.6 | 63.3 | 0 | 0 | 0 | 2.3 | 48.4 |
| T5 | 11B | ✓ | 19.6 | 10.1 | 41.2 | 34.7 | 2.3 | 0 | 0 | 15.9 | 55.8 |

Table 27: Model results on natural language reasoning tasks in the HELM benchmark, with models ordered by their average rank on the tasks. We use "—" when a model was not evaluated on a given metric, or has runtime errors logged in HELM (e.g., "unmapped prediction" for the code-davinci-002 and code-cushman-001 models on LSAT and Legal Support). StarCoder generally substantially outperforms other open-access models, and often outperforms much larger models.

length of one token and a maximum length of 128 tokens. We use nucleus sampling (Holtzman et al., 2020) with $p = 0.95$ to generate all of our responses.

We use two methods for automatically evaluating toxicity in responses: (i) a RoBERTa-based (Liu et al., 2019) toxicity classifier (Vidgen et al., 2021) and (ii) a list of potentially offensive words.[17] For the toxicity detector, we report the percentage of responses flagged toxic using a threshold of 0.5. For the offensive word list, we report the percentage of responses which contain an offensive word. We note that while the offensive word list can potentially falsely flag responses, it may provide a crude measure of blatant toxicity. We report our results in Table 26.

In general, we observe that CodeGen-16B-Multi and StarCoderBase both appear to generate less toxic responses than LLaMA-13B. For instance, 1.43% of LLaMA-13B's responses contain potentially offensive tokens compared to the 1.12% of StarCoderBase. We also note that CodeGen-16B-Multi appears to generate less toxic responses than StarCoderBase.

## 7.4  Reasoning Tasks in HELM

We evaluate StarCoderBase with HELM (Liang et al., 2022), an evaluation suite aiming to increase the transparency of LLMs by reporting their performance on a wide range of tasks. We evaluate the ability of the model to leverage its natural language and code pretraining for natural language *reasoning* tasks from HELM (excluding code tasks, because of our own extensive code evaluations). At the time of writing, the

---

[17] https://github.com/LDNOOBW/List-of-Dirty-Naughty-Obscene-and-Otherwise-Bad-Words

HELM benchmark does not include the CodeGen, CodeGeex, and LLaMA models. Therefore, we compare StarCoderBase with the largest and/or most recent model from each family of "limited" or "open" access models, as classified on the HELM model list,[18] that had been evaluated on a majority of these HELM reasoning tasks as of May 1, 2023. In Table 27 we report the results. We compute each model's ranking on each task, and order models in the table by their average ranking across tasks. StarCoderBase generally obtains substantially stronger performance than all other models with released weights and often performs comparably to or better than much larger models. We speculate that the mixture of code and natural language in the training data contributes to the model's strong performance on these reasoning tasks.

## 8 Qualitative Evaluation

In Appendix E, we highlight several interesting interactions we had with StarCoderBase. We hope these serve as a starting point for researchers and developers interested in further exploring the model's capabilities. We provide examples of how to elicit interesting model behavior using the templates for Git commits, GitHub issues, and Jupyter notebooks in Section E.1. In Section E.2, we demonstrate how to prompt StarCoder to act as a technical assistant without any instruction-tuning. In Section E.3 we find that it is also possible to prompt the model using a combination of meta-data and natural language to obtain higher pass@1 performance on the HumanEval benchmark.

## 9 Attribution Tools

As generative language tools become more ubiquitous and data-intensive, the need to understand and inspect the massive amounts of text they were trained on becomes more pressing, both to understand the failure modes of models as well as provide transparent data governance feedback in the form of attribution tracing and provenance management of a model's generated output. This pressing need for understanding data (Mitchell et al., 2022) is being increasingly recognized and operationalized in the form of dataset inspection tools and toolkits (Akiki et al., 2023; Marone & Van Durme, 2023; Piktus et al., 2023). It is from this vantage point that we are releasing two such data inspection tools: a membership-checking tool and a BM25 search index. These complement the existing "Am I in The Stack" tool which operates at the level of GitHub repository names. The two new tools index only the files used for training and allow for matches on file content. These tools are available as standalone sites but are also integrated into our VSCode demo. This helps users identify parts of the model output that may have been copied from the training data. By utilizing the search index, users can locate the corresponding source file and repository of the copied snippets.

### 9.1 Membership Checking

Marone & Van Durme (2023) propose documenting datasets with membership testing artifacts deemed *Data Portraits*. They provide one specific implementation, based on Bloom Filters (Bloom, 1970), that offers fast and lightweight membership inference. We build a Bloom-filter-based portrait on strings of length 50 characters from the training data. This artifact takes 26 GB, $\sim 3\%$ of the data size. The inference tool is hosted publicly to complement other documentation artifacts. [19]

Generations from the model can be quickly checked to approximately assess the degree of overlap with the training corpus. The VSCode extension supports using this as a rapid, first-pass attribution method. However, this requires that matching strings are longer than a minimum size and does not attempt to filter common or generic code snippets. After the first pass check, users can use the full search index to further assess attribution.

---

[18]https://crfm.stanford.edu/helm/latest/?models=1
[19]http://stack.dataportraits.org/

## 9.2 Search Index

We index the training dataset using Elasticsearch 7.17[20] and provide two search tools to query it: one focused on the Python subset and one covering the entire dataset. The code itself is preprocessed using a lowercase filter and Lucene's `ASCIIFoldingFilter`, tokenized using a 3-gram tokenizer, and indexed using the default Lucene implementation of BM25 as a similarity function. We further index the username and license fields as `keyword` fields allowing for easy filtering and lookup based on these specific metadata fields. Both indexes are currently running in single-node mode on one virtual machine.

# 10 Social Impact and Limitations

## 10.1 Project approach

**Open-science and open-governance** StarCoder is an output of a community research project. The project is conducted in the spirit of Open Science (Woelfle et al., 2011), focused on the responsible development *and* use of Code LLMs. Through open-governance practices conducted throughout the project, priority in decision-making has always yielded to the more responsible option even if this meant introducing limitations that might impact adoption or future research. For example, the Legal, Ethics, Governance Working Group decided to remove and not release a dataset of identified malicious code, even though this data might be useful for future security research.

**Openness and safety risks** Solaiman (2023) explains how the degree of openness in the LLM development process is connected to the potential risks associated with a model release. When systems are developed in a fully closed manner, it is more likely for power to become concentrated among high-resourced organizations, and the small development team may not fully comprehend the impact and long-term consequences of the model being deployed. In addition, closed-development systems are often less auditable by external experts and can impede scientific progress since researchers cannot build upon each other's work. On the other hand, fully open development allows for community research, democratizes access to the models, and enables audits throughout the whole development process. However, without appropriate guardrails, open LLM development poses a higher risk of misuse, as increased model access also increases the likelihood of harm caused by the model. Even though a released API can be shut down, once the model weights are released, it is nearly impossible to retract them. Discussing and implementing responsible AI practices has, therefore, been front and center during the development of our project's LLMs.

## 10.2 Limitations

**Dataset and data licensing** StarCoder was trained on a subset of The Stack v1.2 dataset. This dataset has been filtered using a license detector to only include permissively licensed source code. Nevertheless, the license detector might have incorrectly classified a number of repositories. See Kocetkov et al. (2022) for more details on this license detection process.

**Opt-out process** Although The Stack offers a way to remove developer code, its opt-out process only applies to individual repositories and could benefit from further enhancements. For example, when code is licensed under a permissive or copy-left license, it can be duplicated to another repository, making it challenging to eliminate such copies if the copyright owner chooses to opt out. More work is necessary to create better data control and consent mechanisms for large-scale training sets of LLMs.

**PII detection** Despite our best efforts to remove PII (Section 4), StarCoder may still produce PII (however, note that the model license restricts use that aims to generate or disseminate PII with the purpose of harming others). As mentioned in Section 4.2, we trained an encoder-only model to detect PII for both code- and text-related tasks and noted that there is a possibility of false positives and negatives, which could lead to unintended consequences when processing sensitive data. Moreover, the PII detection model's performance

---

[20]https://www.elastic.co/guide/en/elasticsearch/reference/7.17

may vary across different data types and programming languages, necessitating further validation and fine-tuning for specific use cases. The PII annotations are only available to approved individuals, and researchers and developers who are granted access are expected to uphold ethical standards and data protection measures. By making it accessible, our aim is to encourage further research and development of PII redaction technology.

**Malicious code**  On the Hugging Face platform, where the Stack is hosted, a malicious code detection tool identified 654 files as unsafe. With the help of our community, we removed these files ahead of the release of The Stack v1.2. Nevertheless, The Stack may contain undetected malicious code, and StarCoder might be able to generate malware. The StarCoder OpenRAIL-M license, therefore, includes a use restriction against generating and/or disseminating malware (including — but not limited to — ransomware) or any other content that can be used to harm electronic systems.

**Model limitations**  StarCoder is subject to typical limitations of LLMs, including the potential to generate content that is inaccurate, offensive, misleading, discriminatory towards age or gender, or reinforces other stereotypes. Please refer to Section 7.3 for an investigation into such safety concerns. Deployments of StarCoder need to further challenge and adapt the model to prevent such behavior, e.g., through red-teaming (Perez et al., 2022), adversarial testing (Wan et al., 2023), and/or by adding a robust safety layer (OpenAI, 2023b). The model is released with an OpenRAIL-M license that places enforceable use restrictions that apply to the model and its modifications, and to applications using the model.

**English-only evaluations**  We evaluated the performance of StarCoder solely on English-based benchmarks to understand its coding capabilities and natural language understanding. To make these models more accessible to a wider audience, future research should investigate the performance and limitations of Code LLMs on other natural languages.

**Code attribution tools**  The StarCoder membership-checking tool and BM25 search index are limited to dataset inspection against the subset of The Stack that was used for training and, as such, will not find matches to code that was not included or that was removed from the dataset for this project. The Portraits-based membership testing tool uses hash matching and thus may have false positives. It also has a minimum resolution and requires a certain amount of context to trigger a match. Both attribution tools do not attempt to distinguish between generic code (e.g., boilerplate) or protected content. However, we hope that these tools will support ongoing research on the responsible development of LLMs.

## 10.3 Social impact

**Code LLMs**  We expect Code LLMs to enable people from diverse backgrounds to learn to write higher-quality code and develop low-code applications (Leinonen et al., 2023). Mission-critical software could become easier to maintain as professional developers are guided by code-generating systems on how to write more robust and efficient code. However, the security implications should also be carefully considered (Sandoval et al., 2023). While the social impact is intended to be positive, the increased accessibility of Code LLMs comes with certain risks such as over-reliance on the generated code and long-term effects on the software development job market. We refer the reader to Chen et al. (2021, Section 7) for a broader impact analysis of Code LLMs, as well as Khlaaf et al. (2022) for an in-depth risk assessment and hazard analysis of this emerging technology.

**Data annotation**  It was important for the project to only use reputable data annotation services. It was also important to balance the constraints of costs (fair compensation), time (the timing and time to complete the work were on the critical path for the project), and quality (to ensure that PII Detection Model training was not impacted). While traditional data annotation services using salaried employees were considered, the decision to work with Toloka crowd-workers was taken after a review of service providers and their compensation practices — most would not provide sufficient transparency and guarantees about worker compensation. Our determination of compensation took into consideration different minimum wage rates across countries and their corresponding purchasing power. We limited annotation eligibility to countries

where the hourly pay rate of $7.30 was equivalent to the highest minimum wage in the US ($16.50) in terms of purchasing power parity.

**Feedback opt-out form**    During the first stage of the opt-out process, individuals were asked to specify the reasons for wanting their code to be excluded from the dataset. The recurring concerns we heard from the individual who wished to opt out are:

- Preference for an opt-in approach instead of opt-out.

- Perception that it is unfair to use their code without compensation

- Concerns about the current limitations of AI and the potential for model generations to be traced back to their work, resulting in potential legal liability.

- Belief that their code is of poor quality and unsuitable for AI training.

- Presence of PII in their code, which they do not wish to be publicly exposed.

The opt-out form thus provided an opportunity to directly engage with content creators and learn about the impact of our work on them.

**Community feedback on opt-out process**    We conducted community research with individuals at specific organizations whose data is used in The Stack (The Alan Turing Institute and *The Turing Way*) and contributed to two open, international workshops (Open Data Day 2023 and Mozilla Festival 2023 with a session titled 'Designing for Data Rights in the AI Production Pipeline'). These qualitative interviews and participatory co-design workshops included 50 participants, primarily from North America and Europe, with roles including research scientist, community manager, software engineer, and principal investigator (PI).

The outcomes from the community research can be summarized as follows: when it comes to governance of LLM datasets, participants feel that it is both *better to know* and *better to have a choice*. Most participants had neutral to positive feelings about their permissively licensed data being used to train LLMs. While all had positive impressions of the "Am I in The Stack" tool, not one interviewed expressed a desire to actually opt out. The main takeaway seemed to be that participants found the most value in the project's governance tools for their ability to raise awareness of data practices and to empower individuals and communities to take action based on their specific needs. These initial conversations also highlighted the importance of bringing governance discussions and decisions directly to impacted communities, an important direction of future work that should extend community research beyond North America and Europe. Participants in the workshops also raised examples of new groups to center in data rights considerations, including artists, data miners, and future generations. The co-created outputs can be viewed on this MozFest Miro Board.

## 11    Conclusion

In this technical report, we described the efforts of the BigCode community in creating StarCoderBase and StarCoder, open-access 15.5B parameter large language models trained on code. We provided full transparency on all aspects of the research and development process, including the training data, the data curation process, the PII redaction pipeline, and the model training. We conducted the most extensive evaluation of Code LLMs to date, finding that StarCoder outperforms other Code LLMs like CodeGen (Nijkamp et al., 2023) and CodeGeeX (Zheng et al., 2023), and matches or outperforms the closed-access code-cushman-001 model from OpenAI. By releasing the StarCoder models with an Open Responsible AI Model license, and by open-sourcing all code repositories for building the model on GitHub, we aim to increase access, reproducibility, and transparency of Code LLMs in the research and developer communities. The model license includes use restrictions to ensure that modifications of the model and applications using the model adhere to our principles of responsible AI. In addition, we released a novel set of attribution tools to help end-users of Code LLMs to detect and locate model generations that may have been copied from the training set. We hope these measures contribute towards a safe model release, ensuring that the strong-performing StarCoder models remain a force for good.

**Acknowledgements**  We would thank Hugging Face for providing the compute resources to train the StarCoder models. We also thank Suriya Gunasekar for help with the data inspection, and Sebastien Paquet for proofreading this work. Carolyn Jane Anderson, Arjun Guha, Ming-Ho Yee, and Yangtian Zi and are supported by U.S. National Science Foundation awards SES-2326174 and CCF-2102288. Evgenii Zheltonozhskii is supported by the Adams Fellowships Program of the Israel Academy of Sciences and Humanities.

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
