# OpenReview forum: "StarCoder: may the source be with you!"
_TMLR — Accepted by TMLR_

### Review · Reviewer_s9bJ · 2023-09-04

**Summary Of Contributions:**

This submissions presents StarCoder, a LLM trained on code(-related) data from GitHub. The data collection and filtration strategy is discussed in detail, followed by an extensive evaluation on a number of benchmarks from the literature, showing that StarCoder outperforms earlier similiar-sized models.

**Audience:**

Yes

**Broader Impact Concerns:**

None beyond the ones discussed in the paper.

**Claims And Evidence:**

Yes

**Requested Changes:**

* Please document handling of usernames in PII redaction more clearly, see detailed remarks above.
* Please document FIM more fully, see detailed remarks above.
* Please expand the report to discuss issues observed during training (e.g., where restarts required due to problematic batches?), or clarify that this was just smooth sailing (if yes, congratulations on being extremely lucky!)

**Strengths And Weaknesses:**

Strengths
----
* Clear, mostly complete (apart from oversights, I assume) description of the data collection procedure for a Code LLM.
* Substantial evaluation on a number of different benchmarks, comparing to many relevant Code LLM baselines.
* Accompanies an open-access model release useful for the research community.

Weaknesses
----
* No technical novelty, and no ablations supporting design choices. It's unclear what can be learned from this going forward.

Details
----
* Intro long and somewhat tangential to the core content of the paper (e.g., the paragraph around limits of fair use; I see that it's meant to set the ground for the attribution tool, but maybe it would be better placed)
* Intro includes some (debatable) assertions (e.g., "The Stack is [...] with a [...] ethical data governance framework" is clearly problematic given that there is no exact definition of "ethical", as somewhat indicated by the paper discussing the gap between legal requirements and feasability of obtaining consent at scale)
* Page 8: "Each event includes the author's username" - did you consider removing usernames / replacing them by pseudonyms? (This would be more reliable than relying on PII redaction, which according to Sect 4.3 struggled with this, and have the advantage of being able to preserve identities such as "Author", "Commenter1", "Commenter2", etc.)
* Sect. 3.4/5.1: Can you provide more context on the issues you observed with diff representations? They are a very natural way of avoiding reproduction of unchanged lines. (It would also enable the model to predict changes, rather than new versions)
* Sect. 4 / page 12: "performance for detecting usernames did not show significant improvement, so we decided to exclude it from the PII redaction process." - does this mean that usernames were not redacted at all, or that only the pseudo-label step was skipped for these?
* Sect. 4: did you do an additional study to understand failure modes of the PII detector? In particular, it would be interesting to understand whether it worked better/less well for different kinds of artifacts (e.g., issues vs. code).
* Sect. 5.1: The formatting of templates is inconsistent and not defined (e.g., both "STARS" and "Code" are probably template variables, but that is never defined). Could you make it consistent, briefly explain it, and potentially use color to clarify things?
* Sect. 5.1: Did git commits include filenames? How were multiple files represented?
* Sect. 5.4: I was confused by finding the fill-in-the-middle reference in the model architecture - it's a pure data transformation and hence a better discussed in Sect. 5.1? It would be helpful to learn more about the details here as well (is it applied to all data, or just full code files? PSM/SPM/SPMv2? Ratios? Context/document-level?)
* Sect. 5.5: "we decreased the learning rate to $5 \times 10^{-5}$" is confusing, given that StarCoderBase ends with a LR of $3 \times 10^{-5}$ (i.e., the LR is increased again for the fine-tuning phase).
* Sect. 6.1.1: The highlighting of the comparison to PaLM/LaMDA/LLaMA seems inappropriate given that these are general-purpose models, not specialized to code.
* Sect. 6.1.2: Why did you include SantaCoder as a baseline here, but not in 6.1.1?
* Sect. 6.2.1: "StarCoder shows high performance on Swift, even though it was not included in its training set" - this is a mystery that deserves more explanation. I understand that in the HumanEval-via-MultiPL-E setting, the input should have no signficant context besides the target language. How did StarCoder manage to produce valid Swift code then, given that it did not see it during training time?
* Sect. 6.2.2: Can you explain as to why StarCoder produces more valid / fewer insecure predictions in this setting? It's not clear to me what lesson to draw from the paper here (i.e., what should people do in the future to get similar results).
* Sect. 6.2.3: Infilling of types / single lines can be done naturally by encoder-decoder models trained in the MLM setting, for example CodeT5. Have you considered evaluating against these as well? (For example, CodeT5 outperforms InCoder-6B and SantaCoder on python code summarization)

---

### Review · Reviewer_Zb4V · 2023-09-13

**Summary Of Contributions:**

The paper introduces Starcoder and StarCoderBase, two large language models for code based on the publicly available "The Stack" dataset.  The paper describes the procedure for data preprocessing and training. An evaluation is performed is performed on diverse programming tasks (e.g., code generation, fill-in-the-middle) in different languages. The results show that SatrCoder models are on par or better than the state-of-the-art. Beyond coding, evaluation of the models on reasoning and natural language tasks is also performed.

**Audience:**

Yes

**Broader Impact Concerns:**

The authors have addressed these concerns.

**Claims And Evidence:**

Yes

**Requested Changes:**

* Provide a discussion of the technical differences (if any) over SantaCoder.

* Compare against SantaCoder for the tasks in Tables 12 and 13 or provide justification as to why such a comparison cannot be done.

*  "We use sampling temperature 0.2 for pass@1, and temperature 0.8" for the other models, are the settings the same?

* In 6.1.3, queries in Russian, Japanese, etc. were given to the different code-LLMs. Were the LLMs in the corresponding table trained with text in these languages?

* In 6.2, why were DoP and DoD not considered?

**Strengths And Weaknesses:**

Strengths:

+ The paper tackles an important and popular problem, is well-organized, and is easy to read.

+ The paper evaluates the performance of StarCoder and StarCoderBase on a variety of benchmarks. The results show that StarCoder achieves state-of-the-art performance on many challenging tasks.

+ The models can be a valuable source to the research community.

Weaknesses:

* There is a lot of text describing preprocessing steps which are important for reproducibility but do not represent interesting contributions. In contrast, the text about model training is short. I am left wondering about the intellectual contributions here. Are all the improvements here due to better data or larger token length or are there specific insights in terms of architecture, training, or fine-tuning?

* What are the differences over SantaCoder besides the size of their training data or the number of parameters? Why is SantaCoder not used as a baseline in Tables 12 and 13?

* "Only a handful of recent models support FIM: from OpenAI (Bavarian et al., 2022), InCoder (Fried et al., 2022), and our prior work on
SantaCoder (Ben Allal et al., 2023)" This statement breaks the anonymity of the submission. I have a similar observation for the link in footnote 15. It is possible that this is due to a large number of authors writing different parts of the paper. However, an end-to-end pass must have been done by someone to make sure that the submission stays anonymous.

Other comments:

* "StarCoderBase is also very capable on Python and is only outperformed by the two models that are fine-tuned on Python: CodeGen-16B-Mono and StarCoder itself.", Looking at Table 12, CodeGen-16B-Mono does not outperform StarCoderBase.

* Table 14 is on page 19 and the corresponding text is on page 22, why not move the table down?

---

### Review · Reviewer_dozN · 2023-10-01

**Summary Of Contributions:**

This paper describes StarCoder, an open-source LLM for code. The main focus of the paper is to lay out the design process and decisions, and to conduct extensive benchmarking. The paper uses an existing dataset and an existing architecture (with tweaks) for training StarCoder. It conducts evaluations across multiple benchmarks and gets better results than comparable open and closed models.

**Audience:**

Yes

**Broader Impact Concerns:**

The paper has an impact statement that I feel adequate converts the potential positive and negative impacts.

**Claims And Evidence:**

Yes

**Requested Changes:**

Clarifications and required rectifications:
---
* The paper mentions that USERIDs are detected and excluded as PIIs, but later I find that USERID is part of the format of training examples for GitHub issues data. This seems contradictory.
* The authors cite the Codex paper for pass@k, but state that "a benchmark problem is considered solved if any one of k code samples passes every test case." This is not the definition of pass@k,n in the Codex paper.
* Table 17 gives numbers on samples, it should additionally provide example-wise numbers (how many examples had at least one generation that was valid and secure).

Optional but desirable:
---
* As described in W1, I find that the discussion on the usefulness of the diverse data sources (code, commits and issues) lacking. There is some mention of the last two contributing to NL, but it would be great if the authors can conduct some ablations to probe the contributions of the data sources.

**Strengths And Weaknesses:**

Strengths:
---
S1. It is one of those papers whose contributions cannot be evaluated simply based on technical novelty in terms of modeling or training. First, the final artifact of the paper, the open model, can be valuable to the research community and the authors should be commended for the efforts. Second, the design process and rationales are thought through and well-documented.

S2. The paper describes the end-to-end pipeline nicely: data pipeline (sources, filtering, distribution, crowd sourcing), training (encoder, PII detection and mitigation) and inference (with attribution). I personally found it quite informative.

S3. The paper has conducted extensive evaluation against multiple open and close LLMs on numerous benchmarks and the results show that StarCoder and StarCoderBase are performant.

Weaknesses:
---
W1. As stated above, I fully appreciate the discussion on the end-to-end setup. However, what I found missing was the discussion about the reasons for superior performance of StarCoder compared to baselines. The paper doesn't articulate it clearly but it is likely due to the diversity of data, and data and compute scaling. There is discussion on scaling with evaluation on different checkpoints and correlation to data sizes. I would really like the authors to also inspect the effect of diversity of data.

W2. Despite significant quality control efforts throughout the data pipeline, I was surprised that the authors did not filter for malicious or vulnerable code. There are a number of static analysis tools that can be run at scale, including CodeQL that runs on GitHub.

W3. There is no mention of whether source code of the data and training pipelines of StarCoder are open sourced. I believe they would be equally useful to the research community to design standardized pre-processing (particularly around filtering for PII, data licensing, etc.).

---

### Decision · Action_Editor_jhGz · 2023-11-16

**Recommendation:** Accept as is

**Comment:**

I am recommending acceptance as is since the requested revisions are fairly minor. However, the reviewers did emphasize that they would like to see the requested changes from their reviews applied to the manuscript.

**Audience:**

Yes, this paper presents a suite of open-source models for code generation that are likely to be broadly useful. The paper itself serves as key documentation of this artifact.

**Claims And Evidence:**

The reviewers found the claims of this be paper to be supported by evidence. The paper is clear and presents a coherent, well-documented set of pre-trained models. Most of the comments ask about relatively minor details of the preprocessing, model setup, etc., and are clarified in the rebuttal.